# LARGE LANGUAGE MODELS ARE
# NATURAL VIDEO POPULARITY PREDICTORS

## ABSTRACT

Predicting video popularity is typically formalized as a supervised learning problem, where models classify videos as popular or unpopular. Traditional approaches rely heavily on meta-information and aggregated user engagement data, but video popularity is highly context-dependent, influenced by cultural and social factors that such approaches fail to capture. We argue that Large Language Models (LLMs), with their deep contextual awareness, are well-suited to address these challenges. However, bridging the modality gap between pixel-based video data and token-based LLMs is a key challenge. To address this, we transform frame-level visual data into sequential text representations using Vision-Language Models (VLMs), enabling LLMs to process multimodal video content—titles, frame-based descriptions, and captions—capturing both engagement intensity (view count) and geographic spread (the number of countries where a video trends). Evaluating on 13,639 videos, we show that while a supervised neural network using content embeddings achieved 80% accuracy, our LLM-based method reached 82% without fine-tuning. A combined approach, integrating the neural network's predictions into the LLM, further improved accuracy to 85.5%. Additionally, the LLM generates interpretable hypotheses explaining its predictions based on theoretically sound attributes. Manual validations confirm the quality of these hypotheses and address concerns about hallucinations in the video-to-text conversion process. Our findings highlight that LLMs, equipped with textually transformed multimodal representations, offer a powerful, interpretable, and data-efficient solution to tasks, requiring rich contextual and cultural insights, such as video popularity prediction.

## 1 INTRODUCTION

Video consumption now accounts for the majority of internet traffic and continues to grow rapidly, making video popularity prediction a critical challenge for content creators, social media platforms, and advertisers (Cisco, 2021). Accurately predicting which videos will become popular is not only important for these stakeholders but also provides valuable insights for researchers studying information diffusion (Park et al., 2017; Rezvanian et al., 2023), social influence (Park et al., 2016; Lin et al., 2023b), cultural dynamics (Park et al., 2017; Haldar et al., 2023), and potential misuse in online networks (Beni et al., 2023). Despite its significance, predicting video popularity remains a complex and challenging task due to the wide range of influencing factors, such as historical context (Ng & Taneja, 2023), cultural trends (Park et al., 2017), and even emotional engagement with viewers (Guadagno et al., 2013; Park et al., 2016). The vast diversity and volume of video content online further complicate these challenges.

Despite the growing interest in this area, most existing research has employed statistical and supervised learning approaches that primarily rely on meta-information and aggregated user metrics, such as uploader statistics, view/comment/like counts, and external factors like social network size and early engagement in other platforms (Zhou et al., 2010; Shamma et al., 2011; Borghol et al., 2012; Park et al., 2016). While these factors provide useful signals, they mainly reflect the uploader's reputation or early user reactions and often fail to capture the deeper contextual and cultural significance encoded within the video content. These intrinsic qualities of video content may play a critical role in how a video resonates with both local and global audiences. However, traditional approaches struggle to process and leverage such rich information due to limited technological capacity in processing complex multimodal data.

In this paper, we propose a novel approach to video popularity prediction that shifts the focus to the intrinsic qualities of a video's textual, verbal, *and* visual content, excluding after-the-fact user engagement data such as early view counts and social network signals. Our method leverages the power of Vision-Language Models (VLMs) and Large Language Models (LLMs) to extract and interpret these intrinsic qualities, complemented by conventional descriptors such as titles and descriptions. To address the challenge of integrating pixel-based video data with the token-based architecture of LLMs, we use VLMs to transform frame-level visual data into sequential textual representations. These representations are then combined with conventional video descriptors such as titles, descriptions, and captions (extracted from the video's audio) to create a comprehensive multimodal textual representation. This transformation allows LLMs to perform the video popularity prediction task, effectively capturing both vertical aspects of popularity, such as view counts, and horizontal aspects, such as the global reach of the video across different countries, by utilizing the deep contextual understanding embedded in LLMs.

Empirically, we introduce a prompting strategy that incorporates a novel approach of integrating supervised learning signals into LLM-based predictions. Our method not only outperforms traditional deep learning models based on content embeddings but also provides human-interpretable predictions in the form of attribute-based hypotheses. Additionally, we introduce the *Global Popular Video Dataset (GPVD)*, a large-scale dataset of 1.3M unique popular YouTube videos, enriched with titles, descriptions, and detailed metadata. The dataset uniquely includes three key popularity metrics: view counts, the number of countries where the video trended, and the number of days it remained on trending lists. The latter two metrics, which reflect the video's global reach and sustained popularity, have been overlooked in prior research but are crucial for a comprehensive understanding of video virality. For our experiments, we subsampled this dataset to create a balanced subset that captures different popularity classes, reflecting both engagement intensity and geographic spread, as detailed in the Methods section. Furthermore, we address concerns related to hallucinations in video-to-text conversion and validate the quality of attribute-based hypotheses through survey experiments.

The key contributions of this paper are as follows:

1. We formulate a video popularity prediction task that not only accounts for engagement intensity (e.g., view counts), commonly used in prior work, but also incorporates geographic spread as a critical dimension of popularity.

2. We introduce the *GPVD*, comprising 1.3M representative popular YouTube videos from 109 countries, supplemented with various metadata to support future studies in the field.

3. We propose a framework combining VLMs and LLMs to predict video popularity by transforming multimodal video content into textual representations. This pipeline innovatively integrates supervised learning signals into LLM-based predictions, enhancing performance.

4. We systematically explore and evaluate the impact of different prompting techniques, such as hypothesis generation, KNN-based example retrieval, and supervised signals while also providing human-interpretable predictions validated through systematic human evaluations.

## 2    RELATED WORK

**Video Popularity Prediction**   Video popularity prediction has traditionally been treated as a supervised learning task, focusing primarily on predicting view counts based on factors like title length, runtime, and user engagement (Zhou et al., 2010; West, 2011; Borghol et al., 2012; Wang et al., 2012). Later work introduced more sophisticated methods, such as time-series analysis and user behavior modeling, to track how popularity evolves (Broxton et al., 2013; Pinto et al., 2013; Vallet et al., 2015; Park et al., 2016; Jog et al., 2021). Despite these advancements, video popularity prediction remains challenging, as it demands capturing cultural and social dynamics that traditional approaches, which often focus narrowly on aggregated platform metrics, fail to account for.

Moreover, these studies typically treat video popularity as a unidimensional problem, focusing exclusively on view counts. However, high view counts alone do not fully capture a video's popularity—especially in the era of streaming, where a video with many views may not necessarily be a global hit; it could be localized to specific regions. To address this limitation, we redefine the prediction task by incorporating both engagement intensity (view counts) and geographic spread (global-local reach),

making the task more nuanced and realistic. We argue that LLMs, with their extensive contextual awareness, are uniquely positioned to capture these complexities, encoding cultural and contextual subtleties that previous studies have been unable to leverage.

**Multimodal Learning and Modality Gaps**   Integrating multimodal data for tasks like video understanding has long been challenging, particularly in addressing the modality gap between pixel-based visual data and token-based language models. Existing methods often rely on modular architectures that process visual and textual data independently using pre-trained visual encoders (e.g., ViT) and language models (e.g., BERT), before concatenating their embeddings (Zeng et al., 2022; Alayrac et al., 2022; Li et al., 2023a;b; Lu et al., 2022). While effective to a degree, these approaches fall short of fully capturing complex interactions between visual and textual modalities, particularly the rich temporal contextual information inherent in videos (Chen et al., 2023b; Qin et al., 2023).

Recent advancements have leveraged VLMs to transform video frames into textual representations, enabling LLMs to reason over multimodal content (Bhattacharyya et al., 2023; Khandelwal et al., 2024; Chen et al., 2023a). These approaches utilize pre-trained models and modular pipelines to generate textual summaries of videos, which are then employed for tasks such as classification and user behavior modeling. Collectively, they demonstrate the effectiveness of integrating VLMs with LLMs for video understanding tasks, emphasizing the role of textual intermediaries to bridge the modality gap.

Building on these innovations, our approach introduces a *frame-to-text transformation* pipeline that converts video frames into sequential textual descriptions via VLMs. This enables LLMs to process visual data as richly contextualized text, facilitating unified reasoning across modalities. Compared to modular architectures that merely concatenate embeddings, our method achieves deeper integration of multimodal data, capturing nuanced information from visual and textual content. Moreover, our work extends this pipeline to an end prediction task using systematic prompt engineering, enabling comparisons across prompting techniques and offering insights into performance improvements.

**Natural Language Explanation and Prompt Engineering**   Generating explanations alongside predictions has been shown to enhance both model understanding and performance in complex tasks. Techniques like Chain-of-Thought prompting (Wei et al., 2022) and self-consistency sampling (Wang et al., 2022) have demonstrated how reasoning chains can improve model accuracy while maintaining interpretability. Two-stage approaches, such as hypothesis generation followed by task solving (Wang et al., 2023), suggest that explanations contribute to better performance, but these approaches often add complexity to the prediction pipeline.

Building on the work of Hanu et al. (2023), who demonstrated the use of textual descriptions for multimodal classification, we extend this approach by integrating hypothesis generation directly into the prediction process, reducing the two-stage approach into a more efficient one-step process. Additionally, we incorporate the supervised learning prediction outcomes into the prompt as additional signals to further improve performance. By combining prompt engineering with contextualized inputs from our frame-to-text transformation, we enhance both the prediction accuracy and the generation of interpretable hypotheses, creating a unified and efficient approach to video popularity prediction.

## 3  METHODS

Our approach transforms video content into a sequence of textual descriptions to enable video popularity prediction through LLMs (see Appendix A.1 for an overview of the pipeline and further details). The process involves several key steps. Let $\mathbf{V} = \{f_1, f_2, \ldots, f_M\}$ represent the set of key frames for a video, where $M$ is the total number of frames. Define $\mathbf{C} = \{C_1, C_2, \ldots, C_C\}$ as the set of captions, $T$ as the title, and $D$ as the user-provided description.

We begin with frame-to-text transformations using the *VideoLLava* model (Lin et al., 2023a), which captures the essence of the video content. The output is represented as $S_{\text{VideoLLava}}(\mathbf{V}) = \{S_1, S_2, \ldots, S_S\}$, where $S_i = \text{VideoLLava}(f_i^W)$ and $f_i^W$ represents a frame window around $f_i$. To enhance these textual representations, time-matched captions are integrated, adding contextual detail to ensure each segment of frames is described accurately in both visual and verbal terms.

Finally, our LLM-based method leverages the reasoning capabilities of LLMs to predict video popularity. As part of the pipeline, we also incorporate a prediction outcome from supervised multimodal models, which serves as both a benchmark and a supervised signal for our LLM-based predictions. The following subsections provide a detailed explanation of each component of our approach.

## 3.1 TASK DEFINITION AND DATASET

We redefine the video popularity prediction task by incorporating two key dimensions:

- **Engagement Intensity**: Total view count, reflecting a video's overall reach and audience engagement.
- **Geographic Spread**: The number of countries where the video trends, indicating global reach and resonance.

By incorporating these dimensions, we offer a more nuanced understanding of video popularity. Formally, given a video $v$ and its content features (e.g., frames, audio, captions), the task is to predict a popularity score $p(v) \in [0, 1]$, representing the likelihood of the video being classified as either a 'local hit' or a 'global big hit.'

To support this task, we introduce the *Global Popular Video Dataset (GPVD)*, comprising the top 50 trending videos across 109 countries over 589 days (February 13, 2021, to March 17, 2023). This dataset includes approximately 5,450 observations per day, totaling 3,210,050 observations and 1,302,698 unique videos. Each observation contains the video ID, trending countries, category, title, tags, and popularity metrics (e.g., views, likes, dislikes).[1] Given that the majority of videos neither go viral nor achieve significant consumption levels, gathering a representative sample of globally popular videos is inherently challenging. Unlike prior studies with shorter collection periods (1-5 months) (Park et al., 2017; Ng & Taneja, 2023), our dataset spans over two years with broader geographic coverage. This enables both short-term trend analysis and long-term cross-cultural comparisons. Videos are categorized based on *Engagement Intensity* and *Geographic Spread* into 16 groups using $4 \times 4$ quantiles, with a focus on the two most distinctive classes:

- **Global Big Hit**: Videos in the top 25% for both dimensions.
- **Local Hit**: Videos in the bottom 25% for both dimensions.

This classification enables us to analyze the factors distinguishing globally highly popular videos from those with regional appeal. For our experiments, we randomly selected a balanced sample from the dataset, comprising 6,279 videos classified as 'Local Hit' and 7,360 as 'Global Big Hit.'

## 3.2 TRANSFORMING VIDEO CONTENT INTO TEXT

Our approach for transforming video content into text for LLM-based video popularity prediction involves the following steps:

1. **Frame Extraction**: From the video, we extract a set of evenly spaced frames, $\mathbf{V} = \{f_1, f_2, ..., f_M\}$, selecting 10 frames per minute to ensure adequate content coverage.
2. **Frames to Text Conversion**: The *VideoLLava* model (Lin et al., 2023a) generates descriptive text for each frame. The model, pretrained to capture visual content effectively, outputs contextually relevant descriptions for each frame: $S_{\text{VideoLLava}}(\mathbf{V}) = \{S_1, S_2, ..., S_S\}$, where each $S_i$ is a textual description generated from a window of frames around $f_i$.
3. **Caption Matching and Data Integration**: The sequential frame-based textual descriptions are synchronized with the video's existing captions, aligning visual and verbal elements to improve contextual accuracy. These descriptions, combined with timestamps, are integrated into a cohesive textual narrative that incorporates both visual and temporal details.

---

[1]We will release the dataset, including video IDs, metadata, and Python code for video downloading, in a publicly accessible repository upon the paper's acceptance, ensuring reproducibility and support for future research.

4. **Summarization**: An LLM ($\Phi_{\text{LLM}}$) processes the integrated text $\mathcal{I}$ along with the title $T$ and description $D$ to produce a concise and coherent final summary $\mathcal{F}$. This step ensures the text is polished, removing redundancies while maintaining narrative consistency.

The *VideoLLava* model was selected for its capability to generate high-quality descriptive text from visual content, providing a solid foundation for caption alignment and summarization. This process enhances video content representation, improving the accuracy of our popularity prediction model.

An excerpt of the summarization prompt is shown below:

```
You're an expert in YouTube videos with extensive experience in analyzing
video content and trends.
...
<output>
<scratchpad>
Step 1:  Identify key elements -...  Step 2:  Analyze Segment Transitions -
...
...
</scratchpad>
<complete_summary>

<introduction> Provide an overview of the video's theme and initial setting.
</introduction>
...

</output>
```

### 3.3 VIDEO POPULARITY PREDICTION THROUGH PROMPTING

Our method leverages LLMs for video popularity prediction by employing a structured sequence of prompts. Conceptually, these prompts are grouped into three categories: *context*, *reasoning*, and *transfer*, each enhancing specific aspects of prediction. Starting with a vanilla LLM setup, we progressively incorporate reasoning steps, few-shot learning, near examples, hypothesis generation, and supervised signals to improve performance.

#### 3.3.1 OVERVIEW OF PROMPT COMPONENTS

1. **Context Set**: Establishes the task by including instructions, task definitions, and expected output format, ensuring the LLM understands the prediction objective.

2. **Reasoning Set**: Encourages intermediate reasoning through prompts like "think before evaluating" and "hypothesis generation," enhancing the model's ability to process complex information and provide explanations for its prediction.

3. **Transfer Set**: Refines predictions using knowledge from external examples via techniques such as few-shot learning, near-example selection, and supervised signals.

#### 3.3.2 SEQUENTIAL PROMPTING APPROACH

The prediction task is framed as a four-class classification problem, with classes $c \in \{1, 2, 3, 4\}$ representing increasing levels of popularity, from local hits to global big hits. Classes 1 and 2 correspond to local popularity, while classes 3 and 4 indicate global popularity. For final predictions, we consolidate these into two categories: 'local hits' (classes 1 and 2) and 'global big hits' (classes 3 and 4).[2] The input consists of integrated summaries, titles, and descriptions, denoted as $\mathcal{F}(\mathcal{I}, T, D)$. Prompts were added sequentially to refine performance, as described below:

---

[2]During experiments, we observed that the LLM set a very high threshold for the Global 'Big' Hit class, resulting in a noticeable bias towards the Local Hit class. Introducing buffer classes (1 and 2 for local hits and 3 and 4 for global hits) addressed this issue by (1) allowing the model to express uncertainty through intermediate predictions, avoiding forced binary decisions; (2) creating a more granular classification system that better reflects real-world ambiguities between extreme categories; (3) enabling better-calibrated confidence levels by incorporating buffer zones between classes; and (4) establishing a smoother decision boundary between extremes, reducing the potential for overly rigid classifications. This adjustment significantly enhanced prediction performance, leading us to adopt the four-class structure in our experimentation pipeline. Details of the class setup can be found in Section A.4 of the Appendix, which includes the final prompt used in our experiments.

**Vanilla LLM Prompt (Context Set)** The initial prompt, $\mathcal{P}_{\text{vanilla}}$, includes basic instructions, task definitions, and output format specifications:

$$\mathcal{P}_{\text{vanilla}} = \mathcal{P}_{\text{instructions}} + \mathcal{P}_{\text{task}} + \mathcal{P}_{\text{output}} \tag{1}$$

**Thinking (Context + Reasoning Set)** We added a reasoning step, $\mathcal{P}_{\text{think}}$, encouraging intermediate reasoning as "thinking before evaluating" based on the Chain-of-Thought approach (Wei et al., 2022). This step improved the LLM's ability to interpret information within video content.

**Few-shot Learning (Context + Reasoning + Transfer Set)** Few-shot learning (Brown et al., 2020) was introduced by providing labeled examples $\mathcal{E} = \{(T_i, \mathcal{F}(\mathcal{I}_i, T_i, D_i), y_i)\}_{i=1}^{N}$, where $y_i \in \{1, 2, 3, 4\}$ denotes the popularity class. This enabled the LLM to generalize from analogous examples.

**Near Examples (Context + Reasoning + Transfer Set)** We then incorporated semantically similar $k$ examples, $\mathcal{E}_{\text{near}}$, selected based on cosine similarity between title embeddings generated using the MPNet encoder Song et al. (2020). These examples, $\mathcal{E}_{\text{near}} \subseteq \mathcal{E}$, were added to the prompt: $\mathcal{P}_{\text{near}} = \mathcal{P}_{\text{vanilla}} + \sum_{(T_i, \mathcal{F}(\mathcal{I}i, T_i, D_i), y_i) \in \mathcal{E}_{\text{near}}}$ where $T_i$, $\mathcal{F}(\mathcal{I}_i, T_i, D_i)$, and $y_i$ represent title, full integrated description, and popularity, respectively. This provided the LLM with relevant context to enhance predictions.

**Hypothesis Generation (Full Context + Reasoning + Transfer Set)** Hypothesis generation prompted the LLM to create a set of hypotheses $\mathcal{H} = \{h_j\}_{j=1}^{M}$ based on $\mathcal{E}_{\text{near}}$, using a hypothesis generation function $\Phi_{\text{hypothesis}}$, a prompt designed to produce hypotheses (see Figure A2 for the final prompt). While Wang et al. (2023) adopts a two-stage approach—first generating hypotheses and then solving the task—for inductive reasoning tasks, we streamline the process by integrating hypothesis generation directly into the 'thinking' process. This adjustment enables the LLM to process and synthesize information more effectively, leading to more accurate and interpretable predictions by embedding reasoning within the task-solving step.

**Supervised Signal (Final Prompt)** The final enhancement incorporated supervised signals from a baseline classifier $\mathcal{F}_{\text{classifier}}$, appending information like: "*A supervised model (x% accurate) predicts a popularity rating of {prediction} with {confidence}.*" This signal, though noisy, provided the LLM with an external estimate of the video's potential popularity, thereby encouraging the LLM to weigh external insights as well as to consider the inherent uncertainty in such predictions.

The final prompt, $\mathcal{P}_{\text{Final}}$, was constructed through a straightforward concatenation:

$$\mathcal{P}_{\text{Final}} = \mathcal{P}_{\text{vanilla}} + \mathcal{P}_{\text{think}} + \mathcal{P}_{\text{few-shot}} + \mathcal{P}_{\text{near}} + \mathcal{P}_{\text{hypothesis}} + \mathcal{P}_{\text{supervised}} \tag{2}$$

This comprehensive prompt guided the LLM to integrate reasoning, near examples, and supervised signals for informed predictions.

### 3.4 MANUAL VALIDATIONS OF HALLUCINATIONS AND HYPOTHESIS QUALITY

To evaluate the video-to-text conversion process and the quality of the LLM-generated hypotheses, we conducted two surveys with human evaluators in the US, recruited through Mechanical Turk (MTurk). All participants held at least a Master's degree and were compensated at an hourly rate equivalent to USD 15 for tasks taking approximately 10-15 minutes each. The survey instructions and questions are fully provided in Section A.2 of the Appendix.

For the evaluation, we selected 5 videos from each popularity category (i.e., 5 local hits and 5 global big hits; 10 videos in total), with each video reviewed by 30 independent evaluators. This setup resulted in a total of 300 evaluations for each task: assessing video-to-text conversion and hypothesis and analysis quality. Screening questions were implemented at the end of the survey to ensure high-quality feedback. These questions tested attention to video content, focusing on the video's title, activity, and evaluation metrics. Participants answering all questions correctly were classified as having "passed." Notably, the results showed consistency across both groups—those who passed and those who did not—demonstrating the robustness and reliability of our pipeline's outputs.

### 3.4.1 VALIDATION 1: VIDEO-TO-TEXT CONVERSION QUALITY

The first survey evaluated the accuracy and reliability of the video-to-text conversion process, focusing on potential hallucinations. Participants were tasked with reviewing short video clips and their corresponding model-generated text descriptions. They rated the descriptions on four criteria—accuracy, adherence, consistency, and coverage of the main topic—using a 1-5 scale. The mean ratings from participants are as follows (mean $\pm$ std):

| Metric | All Participants ($N = 30$) | Passed Screening ($N = 12$) |
|---|---|---|
| Accuracy | $4.35 \pm 0.30$ | $4.32 \pm 0.28$ |
| Adherence | $4.28 \pm 0.40$ | $4.22 \pm 0.10$ |
| Consistency | $4.40 \pm 0.25$ | $4.36 \pm 0.22$ |
| Main Topic | $4.56 \pm 0.24$ | $4.55 \pm 0.30$ |

High ratings (all above 4.22) indicate that the text descriptions accurately reflected video content with minimal hallucinations.

### 3.4.2 VALIDATION 2: HYPOTHESIS QUALITY AND LLM ANALYSIS

The second survey assessed the LLM's hypotheses explaining video popularity predictions. Participants rated the quality of the hypotheses and overall analysis on a 1-5 scale (1: Strongly Disagree, 5: Strongly Agree). The mean ratings are summarized below:

| Metric | All Participants ($N = 30$) | Passed Screening ($N = 13$) |
|---|---|---|
| Hypothesis Quality | $4.44 \pm 0.26$ | $4.45 \pm 0.30$ |
| LLM Analysis Quality | $4.15 \pm 0.25$ | $4.24 \pm 0.42$ |

Ratings consistently above 4.15 confirm that the LLM-generated hypotheses were meaningful and the overall analysis of video popularity was of high quality. In short, both surveys validate the accuracy of the video-to-text conversion and the explainability of the LLM's predictions.

## 4 EXPERIMENTS AND RESULTS

**Implementation Details** We used the Claude Sonnet 3.5 (Anthropic AI, 2024), a commercial model, for our experiments, and the LLaMa 3 model (Touvron et al., 2023) for zero-shot and in-context learning tasks. Additional implementation details, including specific configurations for LLaMa, are provided in Appendix A.3. For training the supervised baseline model, we used a learning rate of 0.001 with the Adam optimizer and early stopping (patience=6). See Appendix A.8 for more details.

**Evaluation Metrics** The model's performance was evaluated using accuracy as the primary metric, with precision and recall additionally tracked to assess classification quality.

**Ablation Study** We conducted extensive ablation studies to evaluate the robustness of our model. Specifically, we analyzed the effects of temperature settings, the number of near-examples, and different embedding type choices on performance. Detailed results are presented in Appendix A.6, providing insights into the model's hyperparameter sensitivity and stability across configurations. Furthermore, we extended our analysis to include experiments with Gemini 1.5 Pro, a state-of-the-art Vision-Language Large Model (VLLM). The results, detailed in Appendix A.9, demonstrate consistent performance, further validating the robustness and generalizability of our approach and prompting strategies across diverse architectures.

### 4.1 SUPERVISED MULTIMODAL APPROACH (BASELINE)

As a baseline for video popularity prediction, we implemented a supervised deep learning model that integrates multimodal embeddings from various video features. Textual features such as the title $T$, description $D$, and captions $\mathbf{C}$ are encoded using the MPNet base v2 encoder (Song et al., 2020, $\mathcal{E}_{\texttt{MPNet}} : \mathcal{T} \rightarrow \mathcal{Z}$), generating embeddings $\mathbf{e}_T$, $\mathbf{e}_D$, and $\mathbf{e}C$. Visual features, including video

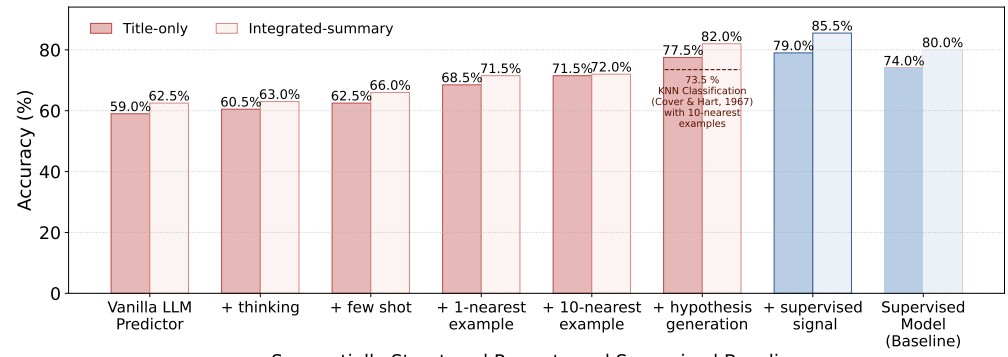

Figure 1: Comparison of accuracy for video-title-only and full-integrated-description-based predictions across models using different prompt sets. The plot shows performance improvements with each model enhancement, beginning with the baseline 'Vanilla' model and culminating in the final configuration incorporating supervised signals. The KNN (Cover & Hart, 1967) and supervised models are included as baselines.

frames $\mathbf{V}$ and thumbnails, are encoded using the CLIP model (Radford et al., 2021; Mendelevitch & Aguynamed, 2023, $\mathcal{E}_{\texttt{CLIP}} : \mathcal{I} \rightarrow \mathcal{Z}$) and its video counterpart ($\mathcal{E}_{\texttt{CLIP-Video}} : \mathcal{V} \rightarrow \mathcal{Z}$) generates frame-level embeddings $\mathbf{e}_{\mathbf{I}_i}$ and a video-level embedding $\mathbf{e}_{\mathbf{V}}$. These multimodal embeddings are concatenated into a unified video representation $\mathbf{v}$, which is fed into a deep neural network for the binary classification ($\mathcal{F}_{\texttt{classifier}} : \mathcal{Z} \rightarrow \{0, 1\}$), predicting whether a video is a 'local hit' or 'global big hit,' $y \in \{0, 1\}$. This baseline model provides a strong benchmark.

## 4.2 Impact of Sequential Prompts on Prediction Performance

The performance of the LLM-based models was evaluated on both title-only and full-integrated-description-based prediction tasks. Figure 1 shows the incremental improvements in accuracy. Starting with the vanilla prompt ($\mathcal{P}_{\text{vanilla}}$), the model achieved 59.0% accuracy for title-based prediction and 62.5% for description-based prediction. The richer information provided in the description resulted in a noticeable improvement in accuracy.

Next, we incorporated additional prompting techniques: thinking prompts ($\mathcal{P}_{\text{think}}$), few-shot examples ($\mathcal{P}_{\text{few-shot}}$), and one- or ten-nearest-example retrieval ($\mathcal{P}_{\text{near}}$), which gradually improved performance, enhancing title-based prediction accuracy from 60.5% to 71.5% and description-based prediction from 63.0% to 72.0%.

The largest gains came from integrating of 10-nearest examples ($\mathcal{P}_{\text{near}}$) and hypothesis generation ($\mathcal{P}_{\text{hypothesis}}$), boosting accuracy by 6.0 and 10.0 percentage points, reaching 77.5% for title-based prediction and 82.0% for integrated description-based prediction. This underscores the value of hypothesis generation, allowing the model to generate and test multiple hypotheses, in enhancing both accuracy and explainability.

The final model, incorporating all components, including supervised signals ($\mathcal{P}_{\text{supervised}}$), achieved 79.0% accuracy for title-based prediction and 85.5% for description-based prediction. This marks improvements of 1.5 and 3.5 percentage points over the previous model and 5.0 and 5.5 percentage points over the traditional supervised baseline, highlighting the value of integrating reasoning and external signals into the prediction process.

Overall, our final model outperformed both well-established supervised methods and baselines utilizing known prompting strategies, such as thinking and few-shot learning. By combining LLM reasoning capabilities with near examples and supervised learning signals, we achieved significant performance improvements, demonstrating the added value of these innovative prompting approaches and highlighting opportunities for further optimization through targeted enhancements and fine-tuning.

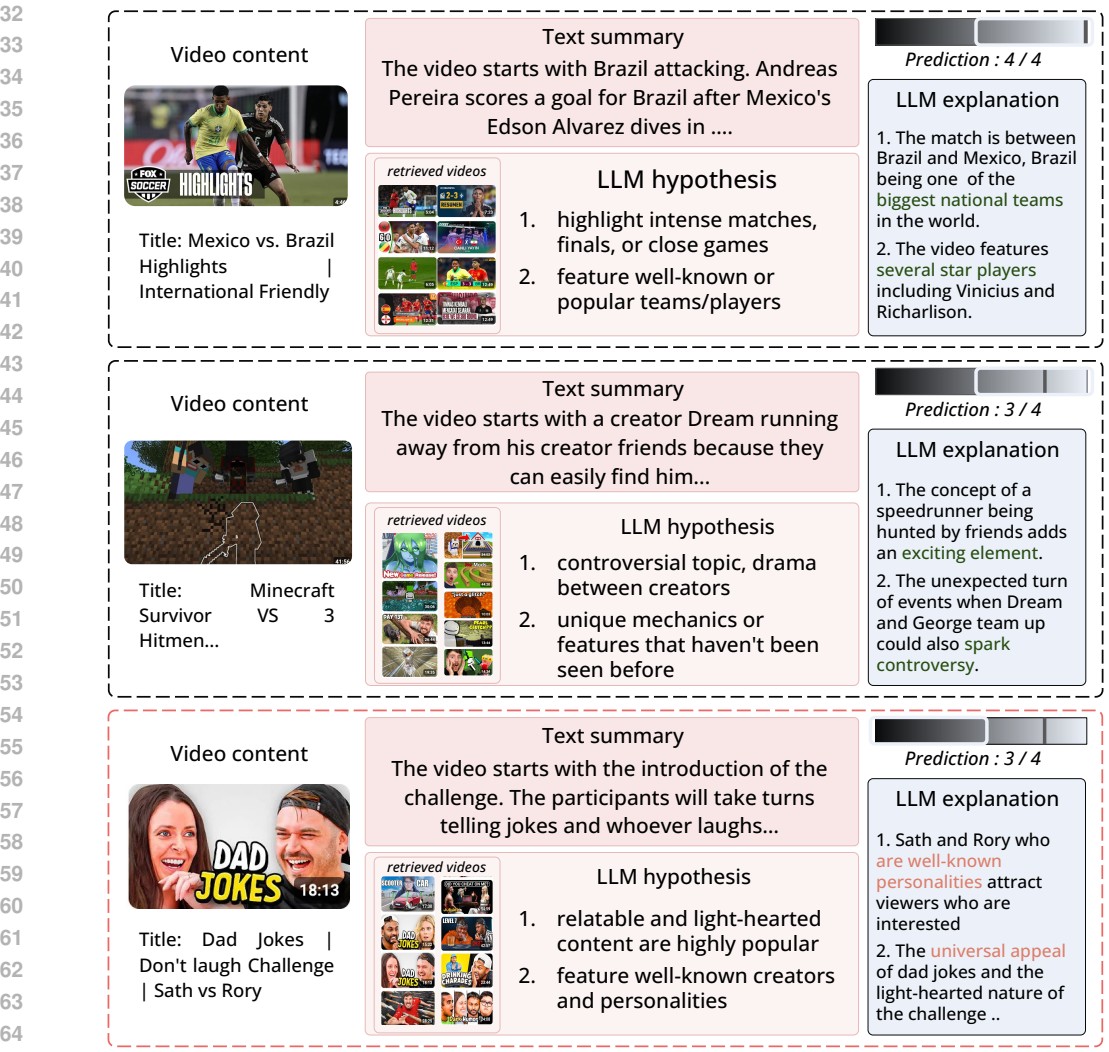

Figure 2: This figure demonstrates the LLM-based framework for explainable video popularity prediction, encompassing text summarization, retrieval of similar videos, hypothesis generation, and final popularity score prediction. It highlights two successful predictions (outlined in black) and one erroneous prediction (outlined in red).

## 4.3 QUALITATIVE EVALUATIONS

Figure 2 presents two successful predictions alongside one incorrect prediction, offering insights into both the strengths and limitations of the approach.

For videos like "Mexico vs. Brazil Highlights" and "Minecraft Survivor VS 3 Hitmen," the framework reasonably identified key drivers of popularity. For the football highlights, the model attributed the video's success to the involvement of Brazil's national team, a globally recognized entity, and star players such as Vinícius and Richarlison. This demonstrates its ability to capture 'geographic spread' by recognizing the global appeal of an internationally celebrated team, as well as 'engagement intensity' through the star power of individual players driving high view counts. Similarly, for the Minecraft video, the model identified the unique mechanics of the speedrun challenge, which appeals to niche gaming communities worldwide, and the engaging personalities involved, reflecting its capacity to detect 'geographic spread' through cross-border relevance in gaming culture and 'engagement intensity' via novel content and influencer-driven popularity. This qualitative analysis highlights how the classification task inherently requires the model to consider both **geographic spread** and

**engagement intensity**. By capturing attributes across these dimensions, the model provides nuanced predictions that align with real-world factors driving video success.

Conversely, the model made an erroneous prediction for the "Dad Jokes" video, revealing certain limitations. While it correctly recognized the universal appeal of humor and the involvement of popular creators, it misjudged the video's overall reach and impact. This highlights the challenge of predicting success for emotionally resonant or culturally specific content, where traditional markers like star power or novelty may not fully apply. These findings underscore the need for further refinement to better capture subtler factors such as sentiment, humor, and cultural resonance, which can transcend conventional indicators of popularity.

## 5 DISCUSSION AND FUTURE WORK

The results of our experiments demonstrate the effectiveness of our approach to video popularity prediction, where LLMs are progressively enhanced through structured prompting techniques. In particular, the performance improvements—from the vanilla prompt to the final model incorporating hypothesis generation and supervised signals—underscore the potential of combining **LLM reasoning capabilities with supervised learning methods**. Notably, **hypothesis generation** not only improved performance as a major contributor to the gains but also enhanced explainability—validated through survey experiments—making predictions more transparent and providing insights into factors driving video popularity.

Our extensive ablation studies, presented in Sections A.6-A.10 of the Appendix, validate the robustness of our framework across various temperature settings, model architectures, and video languages. The results highlight its ability to maintain stable performance under varying conditions, emphasizing its generalizability and reliability even when model parameters fluctuate.

Overall, the results indicate that our approach provides a highly effective solution for multimodal prediction tasks, surpassing traditional supervised models and simpler methods that rely on example-based guidance, such as few-shot and near examples. Additionally, by incorporating two key dimensions of popularity—**engagement intensity** and **geographic spread**—our framework delivers a more nuanced and comprehensive understanding of the factors driving video success, moving beyond the one-dimensional view count focus prevalent in previous research.

Beyond video popularity prediction, the techniques developed in this study have broader implications and applications across various domains. For instance, our VLM-to-LLM pipeline and hypothesis generation methods could generalize to social media analysis, enabling trend, sentiment, or engagement prediction on multimodal platforms. In healthcare, the interpretability of hypotheses generated by LLMs could enhance transparency in medical imaging and diagnosis, where explainable AI is critical for trust and adoption. Similarly, the approach could support education by offering explainable feedback for student assessments or personalized content. Finally, in computational social science, the high-quality hypothesis generation demonstrated by our framework could transform theoretical exploration by offering nuanced explanations, shifting the focus beyond simple statistical coefficients to a richer understanding of sociological and cultural phenomena. These broader applications underscore the versatility and transformative potential of our approach.

Future research could explore several promising directions. Refining the hypothesis generation process, particularly by incorporating advanced reinforcement learning techniques, holds potential for further accuracy improvements. Specifically, the LLM can act as an agent generating hypotheses about video popularity factors, with each hypothesis representing an action within the state space of possible predictions. Prediction accuracy serves as a reward signal, guiding the system to learn which hypotheses are most effective. Additionally, improving the model's ability to account for cultural and emotional factors could enhance predictions for content like humor or emotionally resonant videos, where traditional metrics (e.g., view counts) may be insufficient. A deeper understanding of these factors could also enable the model to better align with user contexts, gauging video appeal based on situational factors and intent. Further integration of multimodal data, such as audio analysis or granular sentiment analysis of comments, could offer richer insights into the drivers of video popularity. Finally, future work could explore real-time prediction capabilities and adapt this framework to predict trends across platforms beyond YouTube.

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

# A  APPENDIX

The appendix provides an in-depth exploration of our video popularity prediction framework, offering detailed analyses and insights to supplement the main text. Its primary objectives are to demonstrate the robustness of our approach, provide a comprehensive understanding of the dataset, and highlight the performance improvements achieved over baseline models. Additionally, we present key ablation studies to examine the impact of various hyperparameters on our model's performance.

## A.1  OVERVIEW OF PIPELINE

Figure A1 presents an overview of our approach. This high-level view depicts our training-free framework for video popularity prediction, which leverages modality-aligned Vision-Language Models (VLMs) and Large Language Models (LLMs) to generate *Video as Text* summaries. In the preprocessing stage (**left**), video content is transformed into sequential text representations using VLMs. During the content aggregation stage, visual and textual information is aligned and combined. The LLM then processes these text summaries to predict a video's popularity score (ranging from 1 to 4) and generates explanations based on identified patterns (**right**). These explanations take the form of hypotheses grounded in theoretically sound attributes. For example, if the video is about a national football game organized by FIFA, the model may highlight its global appeal due to the prominence of the organization and the attention drawn by specific teams, such as Brazil.

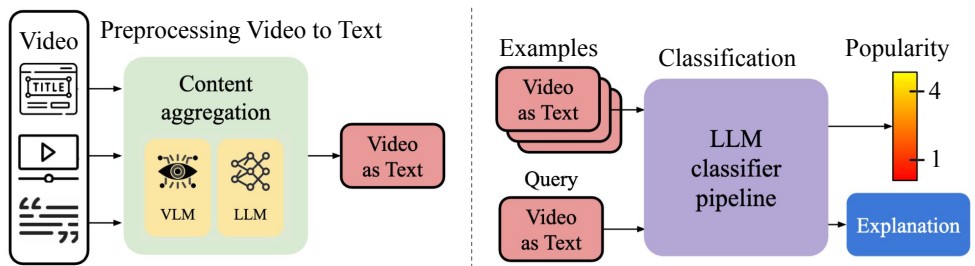

Figure A1: A training-free framework for video popularity prediction utilizing modality-aligned VLMs and LLMs. The **left** shows the video preprocessing and content aggregation stages, where video content is transformed into sequential text representations through VLMs. This transformation generates *Video as Text* summaries, combining visual and textual information. The **right** illustrates the classification and prediction stages, where the LLM processes the *Video as Text* summaries to predict a popularity score and provides an explanation based on the identified patterns.

## A.2  SURVEYS FOR HUMAN EVALUATION

To evaluate the video-to-text conversion process and the quality of LLM-generated hypotheses, we conducted two separate surveys. The first focused on assessing video-to-text conversion quality, while the second evaluated the quality of hypotheses and LLM analysis. Each survey included clear instructions, detailed evaluation criteria, and screening questions to ensure participant attention and understanding. Participants were recruited through Amazon Mechanical Turk (MTurk) and compensated at an hourly rate equivalent to USD 15, reflecting fair pay for tasks requiring approximately 10–15 minutes each.

### A.2.1  SURVEY 1: VIDEO-TO-TEXT CONVERSION QUALITY

> **Video-to-Text Conversion Quality: Initial Instructions**
>
> Welcome to our research study on video-to-text transcription quality. We are academic researchers from ****, investigating the accuracy of automated transcription systems.

## SURVEY OVERVIEW

In this survey, you will:

- Watch short video clips
- Read the corresponding automated transcriptions
- Answer questions about the accuracy and quality of the transcriptions

The survey should take approximately 8-12 minutes to complete. You will receive:

- ☐ Base compensation: USD 1.25
- ☐ Potential bonus: Up to USD 5 total for high-quality responses

## YOUR ROLE AS AN EVALUATOR

Describing video content accurately is an incredibly complex task for AI. It requires understanding context, nuance, and implied information—skills that come naturally to humans but are extremely challenging for machines. Your task will involve:

- Watching diverse video clips
- Reviewing AI-generated content descriptions
- Providing detailed feedback on accuracy and quality

## KEY EVALUATION AREAS

When assessing the AI-generated descriptions, please consider:

- Overall accuracy in capturing key themes and concepts
- AI's ability to understand context and implied information
- Areas where the AI shows particular understanding
- Opportunities for improvement

## IMPORTANT CONSIDERATIONS

Please note:

- The AI system provides a comprehensive overview, not word-for-word transcription
- Focus on overall meaning and key points rather than exact phrasing
- The AI may make contextual inferences

## READY TO BEGIN?

☐ I understand all the above instructions thoroughly

---

### Video-to-Text Conversion Quality: Evaluation Criteria

#### 1. OVERALL ACCURACY
How accurately does the text description match the content of the video?

- ☐ 1: Completely inaccurate
- ☐ 2: Mostly inaccurate
- ☐ 3: Somewhat accurate
- ☐ 4: Mostly accurate
- ☐ 5: Highly accurate

#### 2. CONTENT ACCURACY
How closely does the text description stick to the content presented in the video?

☐ 1: Mostly unrelated to video content

☐ 2: Significant deviations from video content

☐ 3: Moderate adherence to video content

☐ 4: Close adherence to video content

☐ 5: Perfectly matches video content

### 3. MAIN TOPIC CAPTURE

How well does the text description capture the main topic(s) discussed in the video?

☐ 1: Misses all main topics

☐ 2: Captures few main topics

☐ 3: Captures some main topics

☐ 4: Captures most main topics

☐ 5: Accurately captures all main topics

### 4. KEY POINT COVERAGE

To what extent are key points from the video included in the text description?

☐ 1: Misses all key points

☐ 2: Includes few key points

☐ 3: Includes some key points

☐ 4: Covers most key points

☐ 5: Covers all key points

---

### Video-to-Text Conversion Quality: Task: Video Transcription Evaluation

Watch this Video, you will be given the task of evaluating a short transcript about this video next:

Video title: Cristiano Ronaldo Hat-Trick! | Manchester United 3-2 Norwich | Highlights

[Youtube Video]

#### Important Note

Your careful attention to this video is essential for accurately understanding the issues related to AI behavior and the quality of AI-generated video transcriptions. The more accurately you understand and remember the video's content, the more accurate and valuable your evaluation will be. We encourage you to watch the entire video attentively, as your insights will directly impact the assessment of AI performance! Participants who demonstrate a thorough understanding of the video content will be eligible for bonus compensation. Thank you for your dedication to this task!

### TRANSCRIPT

#### INTRODUCTION

This video is a recording of a football match between two teams, featuring commentary and analysis throughout. The initial setting is a stadium with players from both teams on the field.

SEGMENT DETAILS

- **Segment 1:** The first segment shows a man walking on the field while another man walks in the background. The commentator describes the scene, mentioning the ball and a goal scored by Adrian Luna.
- **Segment 2:** In this segment, a man is seen walking towards the camera, wearing a yellow shirt, while another man sits on the ground, wearing a red shirt. The commentators discuss the game, highlighting great deliveries and goals scored.
- **Segment 3:** This segment shows more gameplay, with the commentators analyzing the players' moves and discussing the score.

OVERALL

The overall impact of this video is an immersive and engaging experience for football fans. The combination of exciting commentary, intense gameplay, and skilled players creates a thrilling narrative that will likely appeal to viewers who enjoy sports content.

| EVALUATION | 1 Not at all | 2 Slightly | 3 Somewhat | 4 Mostly | 5 Completely |
|---|---|---|---|---|---|
| How accurately does the text description match the content of the video? | ☐ | ☐ | ☐ | ☐ | ☐ |
| How closely does the text description stick to the content presented in the video? | ☐ | ☐ | ☐ | ☐ | ☐ |
| How consistent is the information in the text description with the facts presented in the video? | ☐ | ☐ | ☐ | ☐ | ☐ |
| How well does the text description capture the main topic(s) discussed in the video? | ☐ | ☐ | ☐ | ☐ | ☐ |

### Video-to-Text Conversion Quality: Screening Questions

**Q1.** Which videos did you evaluate in this survey?
- ☐ SUPAHOTFIRE vs BLUEFACE
- ☐ Cristiano Ronaldo Hat-Trick
- ☐ Kerala Blasters FC vs Jamshedpur FC Highlights
- ☐ Where I'm Travelling Next- Solo Trip?

**Q2.** In the videos you evaluated, which of the following activities was *not* mentioned?
- ☐ Playing rock-paper-scissors
- ☐ Eating cake
- ☐ A football match
- ☐ Skydiving

**Q3.** In the evaluation process, what were you asked to rate about the video transcriptions?
- ☐ Overall Accuracy
- ☐ Content Accuracy
- ☐ Main Topic Capture
- ☐ Video Production Quality
- ☐ Key Point Coverage

**Q4.** Thank you for participating in our study. Your insights will help us improve our AI model for predicting video popularity. Do you have any additional comments or feedback about the model's predictions or this survey?

### A.2.2 SURVEY 2: HYPOTHESIS QUALITY AND LLM ANALYSIS

---

Hypothesis Quality and LLM Analysis: Initial Instructions

#### WELCOME

Welcome to our research study on video popularity prediction using AI models. We are academic researchers from $\cdots$, investigating how Large Language Models (LLMs) can predict video popularity.

#### SURVEY OVERVIEW

In this survey, you will:
- Read video descriptions and LLM-generated hypotheses about video popularity
- Rate the accuracy and relevance of these hypotheses
- Assess the LLM's ability on critical analysis and judgment

The survey should take approximately 8-12 minutes to complete. You will receive:
- ☐ Base compensation: USD 1.25
- ☐ Potential bonus: Up to USD 5 total for high-quality responses

#### STUDY OVERVIEW

We are evaluating a Language Learning Model (LLM) designed to predict video popularity based on content analysis. The LLM analyzes videos from YouTube's trending page and generates hypotheses about what makes videos popular, as well as providing a detailed analysis of each video's content. Your role is to:
- Rate the hypotheses generated by the LLM
- Assess the LLM's critical analysis and judgment for TWO separate videos

#### VIDEO POPULARITY RATING SCALE

The LLM rates videos on a 4-point scale:
- **Popular**: Likely to have general appeal and be popular for a short while
- **Moderately Popular**: Has several appealing elements for more than basic popularity
- **Highly Popular**: Likely to be popular among a broad audience but may not reach ultra popularity
- **Ultra Popular**: Strong potential to become ultra popular, featuring unique, engaging, and broadly appealing content

*Note: While the LLM uses this 4-point scale to rate video popularity, your task will be to rate your agreement with the LLM's hypotheses and analysis using a different 4-point scale.*

---

Hypothesis Quality and LLM Analysis: Instructions - Part 2

#### YOUR TASKS

##### TASK 1: RATE THE LLM'S HYPOTHESES

You will be presented with video descriptions and the LLM's hypotheses about what makes them popular. You will rate your agreement with 4-5 specific hypotheses generated by the LLM about what makes the video popular. For example, a hypothesis looks like '*Sport highlights, especially from important matches, tend to be ultra popular*', to which you can Strongly agree, Agree, Disagree or Strongly Disagree. Your job is to rate how much you generally agree or disagree with given hypothesis based on the same information provided to the LLM.

##### TASK 2: ASSESS THE LLM'S CRITICAL ANALYSIS

You will evaluate the LLM's critical analysis of the video, including its assessment of factors influencing popularity and its final popularity prediction.

## RATING SCALE FOR YOUR RESPONSES

You will use a 4-point scale to rate both the LLM's hypotheses and its critical analysis. This scale is designed to encourage you to form a definitive opinion based on your knowledge and the information provided. For both tasks, use the following scale:

- **1 - Strongly Disagree**: The hypothesis or analysis is clearly incorrect or irrelevant
- **2 - Disagree**: The hypothesis or analysis has major flaws or inaccuracies
- **3 - Agree**: The hypothesis or analysis is mostly accurate and relevant
- **4 - Strongly Agree**: The hypothesis or analysis is highly accurate and insightful

## TIPS FOR COMPLETING THE TASKS

- Read each video description and LLM hypothesis carefully before rating
- Consider each hypothesis and analysis point carefully. Draw on your own knowledge of popular online content, but focus primarily on the information provided in the video description
- For instance, if you think the LLM's hypothesis about sports highlights is accurate based on the video description and your knowledge, you might select 'Agree' or 'Strongly Agree'
- Try to be consistent in your ratings across similar types of content

Your thoughtful evaluations will help us improve the LLM's ability to predict video popularity, ultimately contributing to a better understanding of content trends on platforms like YouTube.

---

### Hypothesis Quality and LLM Analysis: Video: Minecraft but there's Cartoon Hearts

> **Video summary**
>
> Introduction: The video opens with a mysterious green figure walking around a dark room, setting the tone for a fantastical and humorous adventure. Segment Details - Segment 1: Introduces the green figure, referencing Shrek and showcasing magic powers · · · Conclusion: The video's impact is significant, as it showcases the creators imagination and ability to blend disparate elements into a cohesive narrative. The humor, entertainment value, and references to popular franchises will likely appeal to viewers who enjoy fantasy, scifi, and comedy.

In the next two pages, you will be shown 2 tasks:

- Task 1: Rate the hypotheses generated by the LLM
- Task 2: Assess the LLM's critical analysis and judgment

## TASK 1: RATE THE LLM'S HYPOTHESES

### VIDEO INFORMATION

> **Important Note**
>
> Your careful attention to this video description is essential for accurately understanding the quality of AI-generated hypothesis. The more accurately you understand the video's content, the more accurate and valuable your evaluation will be. We encourage you to read the entire video description attentively, as your insights will directly impact the assessment of AI performance!
> Participants who demonstrate a thorough understanding of the video content will be eligible for bonus compensation. Thank you for your dedication to this task!

## MODEL'S HYPOTHESES

| | Strongly Disagree | Disagree | Agree | Strongly Agree |
|---|---|---|---|---|
| H1: Videos with unique Minecraft concepts tend to be ultra-popular | ☐ | ☐ | ☐ | ☐ |
| H2: Content that blends multiple franchises or pop culture elements has broader appeal | ☐ | ☐ | ☐ | ☐ |
| H3: Videos with humorous and imaginative content encourage sharing and discussion | ☐ | ☐ | ☐ | ☐ |
| H4: Fast-paced content with diverse visual elements keeps viewers more engaged | ☐ | ☐ | ☐ | ☐ |

If you Disagree/Strongly Disagree, do you have any better hypotheses or suggestions for improving the provided hypotheses?

## TASK 2: ASSESS THE LLM'S CRITICAL ANALYSIS

### FACTORS IN THE GIVEN VIDEO THAT COULD INFLUENCE POPULARITY:

- F1: Creative concept: "Minecraft but there's Cartoon Hearts" (very positive)
- F2: Blending of multiple franchises (Shrek, Teen Titans, Star Wars, Scooby-Doo) (positive)
- F3: Humorous and playful tone (positive)
- F4: Imaginative scenarios (magic powers, cyber crystals, outer space) (positive)
- F5: Alignment with geek culture trends (positive)
- F6: Potential for viewer engagement and discussion (positive)

**LLM's Final Analysis:** Considering all factors, especially the similarity to other ultra popular videos and the supervised model prediction, this video is likely to be Ultra Popular. It has all the elements of highly engaging content that tends to perform exceptionally well, particularly in the gaming and geek culture niches.

| | Strongly Disagree | Disagree | Agree | Strongly Agree |
|---|---|---|---|---|
| F1: Creative concept | ☐ | ☐ | ☐ | ☐ |
| F2: Blending of multiple franchises | ☐ | ☐ | ☐ | ☐ |
| F3: Humorous and playful tone | ☐ | ☐ | ☐ | ☐ |
| F4: Imaginative scenarios | ☐ | ☐ | ☐ | ☐ |
| F5: Alignment with geek culture trends | ☐ | ☐ | ☐ | ☐ |
| F6: Potential for viewer engagement | ☐ | ☐ | ☐ | ☐ |

### Hypothesis Quality and LLM Analysis: Screening Questions

**Q1.** What was the primary task you were asked to perform in this survey?

- ☐ Predict which videos would become viral
- ☐ Assess the LLM's hypotheses and analysis about video popularity
- ☐ Provide your own theories about what makes videos popular
- ☐ Compare different AI models' performance in analyzing videos

> These video-specific screening questions were randomly assigned based on the viewed video:

**Q2.** Which of the following elements were present in the video you analyzed? (select all that apply)

- ☐ Commentary and analysis
- ☐ Great deliveries and goals scored
- ☐ Interviews with team managers

□ Slow-motion replays
□ Adrian Luna scoring a goal
□ Penalty shootout

**Q3.** Which of the following elements were present in the video you analyzed? (select all that apply)

□ References to Shrek
□ Mining of cyber crystals
□ Outer space scenes
□ Pokémon battles
□ Teen Titans-inspired content
□ Underwater exploration

**Q4.** Which of the following elements were present in the video you analyzed? (select all that apply)

□ Internal struggle of the protagonist
□ Car chase scenes
□ Self-harm depicted
□ Comedic dialogue
□ Apology and plea for forgiveness
□ Transformation sequences

**Q5.** Which of the following elements were present in the video you analyzed? (select all that apply)

□ Two men playing video games
□ Reaction to a music video
□ Wearing headphones
□ Dancing performances
□ Occasional singing into microphones
□ Cooking demonstrations

## A.3 IMPLEMENTATION DETAILS

The video popularity prediction pipeline was implemented using PyTorch 2.1.0 and the transformers 4.35.0 library. The content aggregation was performed using a custom module that combined textual information from titles, descriptions, and generated captions. We employed Claude 3.5 Sonnet (accessed via Anthropic's API) (Anthropic AI, 2024) as our primary LLM for classification and hypothesis generation, while also using LLaMa 3 70B (Touvron et al., 2023) Instruct offline on two NVIDIA RTX A4000 GPUs using tensor parallelism and quantization (2.8 bits per weight) for efficient video-to-text generations. The VLM component (of VideoLLava) used a fine-tuned CLIP model to align visual and textual features. The entire pipeline was orchestrated using a custom Python script that handled data flow between components, with batching implemented to optimize throughput. For the offline LLaMa 3 70B setup, we achieved approximately 2 tokens per second inference speed. All experiments maintained consistent hyperparameters (Temperature: 0.5, Max Tokens: 4096, Top-p: 0.95) to ensure reproducibility.

## A.4 FINAL PROMPT

Figure A2 illustrates the final prompt structure for video popularity prediction using LLMs. The prompt incorporates step-by-step instructions, contextual elements, and a structured output format designed to guide the LLM in analyzing and predicting video popularity.

Key features of the prompt include:

- Instructions: Clear definitions of the task and the four popularity classes, ranging from "Locally Moderately Popular" to "Globally Ultra Popular," to ensure the model understands the classification criteria.
- Step-by-Step Reasoning: A scratchpad mechanism encourages the model to reason through intermediate steps, such as comparing the given video to similar examples, considering supervised model predictions, and generating hypotheses to explain observed patterns.
- Structured Output: The prompt specifies a coherent format for outputs, including an evaluation rating and explanatory reasoning, ensuring interpretability and consistency.

---

**Final Prompt for Video Popularity Prediction**

<Instructions>
You will be predicting the potential popularity of a video based on its title and a description of its content.
Note: All videos in this dataset are from YouTube's trending page, meaning they have already achieved a significant level of popularity.
Your task is to provide a 'popularity rating' indicating how likely the video is to become popular among viewers, using the following 1-4 scale:
1 - Locally Moderately Popular: The video is likely to appeal to be popular, and has elements of general appeal and is probable to be popular for shorter while.
2 - Locally Popular: The video has several appealing elements for more than basic popularity.
3 - Globally Highly Popular: The video is likely to be popular among a broad audience but may not have elements that lead to ultra popularity status.
4 - Globally Ultra Popular: The video has strong potential to become ultra popular, featuring unique, engaging, and broadly appealing content.
</Instructions>

- - - - - - - - - - - - - - - - - - - - - - - - - - - - - - - - - - - - - - - - - - - - - - - - - - -

<output>
<scratchpad>
Think step by step inside <scratchpad>Your analysis here</scratchpad>.
Step 1: Look at the given <video_description>{description}</video_description>, and First, answer the question: "comparing this video with videos in <similar_examples>, are videos similar to this video in the popular or ultra popular category?".
Step 1.5: A supervised model (80% accurate) predicts a popularity rating of {supervised_prediction} with {supervised_confidence:.2f} confidence. Factor this into your analysis.
Step 2: Then create 4 hypothesis about why videos in the <similar_examples>are ultra popular (Evaluation=4) and some popular (Evaluation=1). Try to catch patterns from these example videos, try to generalise patterns that make a video reach high popularity, and why some stay in basic popularity.
...
Step 4: Now expand on that reasoning think about whether the given and <video_description>are going to be Locally Moderately Popular, Locally Popular, Globally Highly Popular, or Globally Ultra Popular, give more weight to answer of step 1: 1 is for "Locally Moderately Popular" and 4 is for "Globally Ultra Popular."
</scratchpad>
<Evaluation>rating</Evaluation>
</output>

- - - - - - - - - - - - - - - - - - - - - - - - - - - - - - - - - - - - - - - - - - - - - - - - - - -

Now predict the popularity for the given video title and description, using the proper output format. Your insights could help shape the future of video content creation.

Figure A2: The final prompt structure for video popularity prediction using Large Language Models. The prompt incorporates instructions, a structured output format, and a step-by-step analysis process.

## A.5 DATASET ANALYSIS

Our dataset analysis uncovers several key insights that help contextualize the dynamics of video popularity and inform our prediction task design. We highlight the importance of considering both *geographical reach* and *view count* as critical factors in assessing a video's popularity. Our analysis shows a positive correlation between a video's international presence and its view count, with notable clusters of videos distributed across different quadrants of this relationship.

Additionally, a three-dimensional analysis that includes the duration a video remains on trending lists reveals a more nuanced relationship between trending duration, geographical reach, and view counts.

We observe an optimal range of 100-200 units of trending duration and a reach of 15-35 countries as indicators of peak performance. However, outliers in this analysis suggest that content quality and other unquantified factors play important roles in determining a video's success.

### A.5.1 DATASET FEATURES AND VIDEO CATEGORIZATION

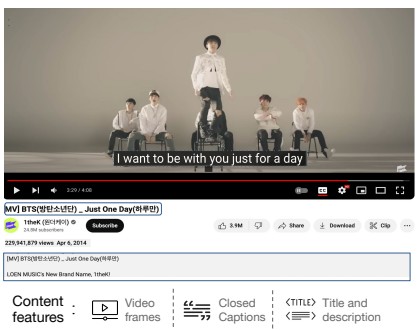
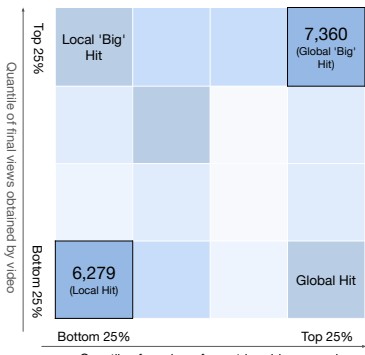

Figure A3: **Left:** The list of features for a youtube video. **Right:** Heatmap categorization of YouTube videos into 16 quantiles based on two key dimensions: the number of views and the number of countries in which the video trended. Videos are classified into 'Global Big Hit' (top 25% in both dimensions) and 'Local Hit' (bottom 25% in both dimensions), with cell colors indicating the relative density of each class.

We present the features used for each video (Figure A3, left) and a heatmap categorizing videos based on engagement intensity and geographical reach (Figure A3, right). This information provides crucial context for understanding the nature of our dataset and how we distinguish globally viral videos from those with more localized popularity.

### A.5.2 GEOGRAPHICAL REACH VS. VIEW COUNT

To better understand how a video's international presence relates to its popularity, we analyzed our dataset, focusing on the connection between the number of countries a video reaches and its total views. Figure A4 visualizes this relationship.

The plot is bisected by two red lines representing the median values for each dimension, effectively partitioning the data into four distinct quadrants. The median number of countries reached by a video is 47.5, while the median total view count is approximately 2.9 million (2,896,886 views). The visualization shows that videos reaching more countries tend to get more views, but not all videos are spread out evenly. We observe a general positive correlation between a video's geographical reach and its view count, suggesting that videos with broader international appeal tend to accumulate more views. Also, a significant cluster of videos is concentrated in the lower-left quadrant, indicating a substantial number of videos with both limited geographical reach and relatively low view counts. While the upper-right quadrant contains videos that have achieved both high view counts and extensive geographical reach, representing the most globally popular content in our dataset.

### A.5.3 THREE-DIMENSIONAL ANALYSIS: TRENDING LISTS, REACH, AND VIEW COUNTS

We conducted a three-dimensional analysis that examines the relationship between the duration a video remains on trending lists, its geographical reach, and its total view count. This analysis, visualized through a heatmap (Figure A5), provides a more comprehensive understanding of the complex dynamics of spread.

The heatmap reveals that videos with longer trending durations and broader international reach generally garner higher viewership, though the relationship is not entirely linear. Peak viewership, depicted by the darkest regions on the heatmap, is concentrated in the upper-right quadrant. This suggests that videos with extended sequences (roughly 150–200 units) and significant global reach (spanning 30–40 countries) tend to achieve the highest view counts. The single highest point, with

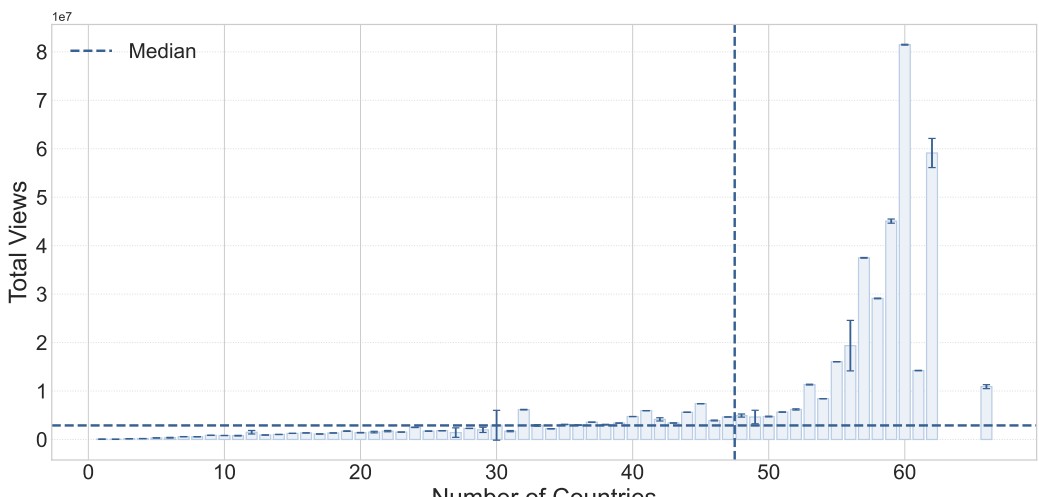

Figure A4: The plot depicts the relationship between the number of countries a video reaches and its total view count. Blue lines represent the median values for each dimension, dividing the plot into quadrants.

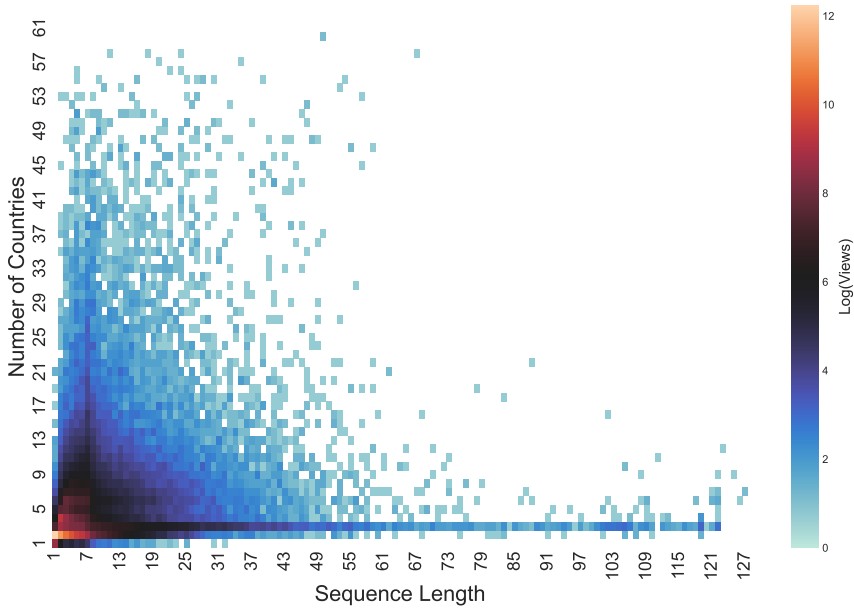

Figure A5: Heatmap depicting the relationship between trending duration (x-axis), number of countries reached (y-axis), and the logarithm of total views (color intensity) (say that image is truncated to 127 sequence length).

a logarithmic view count of approximately 12.5, corresponds to a sequence length of 175 and an international presence in 35 countries.

Examining sequence length patterns, we observe a notable increase in viewership as sequence length increases from 0 to about 100 units. Beyond this point, the relationship becomes more complex, with videos between 100-200 units performing particularly well, especially when they reach a moderate to high number of countries. Interestingly, there's a slight decline in viewership for extremely long sequences (200+), suggesting an optimal range for sequence length.

The impact of country reach on viewership is evident, with videos reaching more countries generally receiving more views. However, this relationship varies across different sequence lengths. For shorter sequences (0-50), the impact of reaching more countries is less pronounced, while it becomes more significant for medium to long sequences.

Several notable patterns emerge from this analysis:

- A clear region of low viewership is visible in the bottom-left corner, corresponding to short sequences with minimal international reach.

- A 'hot zone' appears in the middle-right area of the heatmap, encompassing sequence lengths of 100–175 and country counts of 15–35, where viewership is consistently high.

- Evidence of potential diminishing returns is observed for sequences exceeding 200 in length and country counts above 40.

- Scattered 'hot spots' are present across the heatmap, highlighting outlier videos with unexpectedly high viewership.

### A.5.4 VIDEO CATEGORIES

YouTube's content ecosystem is diverse, encompassing a wide range of video categories. Our dataset provides a unique opportunity to analyze popularity trends across these categories, offering insights that are typically challenging to obtain. In this section, we present an analysis based on 11 heatmaps, each representing a distinct YouTube category (Figure A6). These heat maps visualize the complex interaction between the length of the sequence, the number of countries reached, and the total views for each category. This multi-dimensional analysis reveals both overarching trends and category-specific patterns in video popularity.

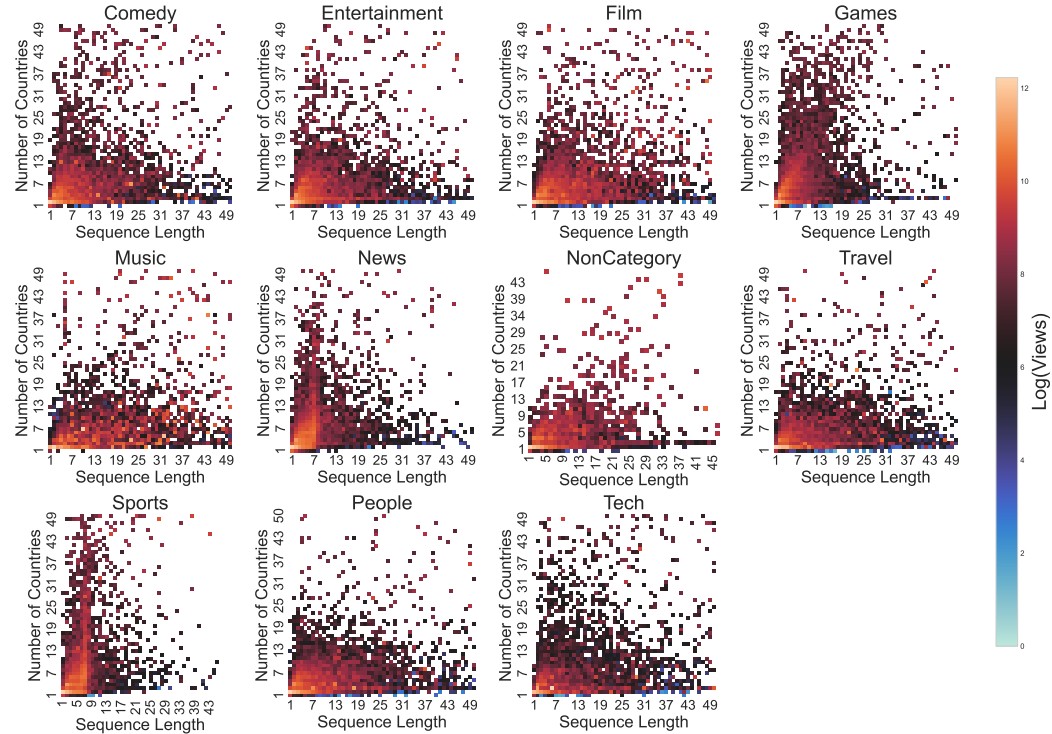

Figure A6: Heatmaps depicting the relationship between sequence length, number of countries reached, and logarithm of total views for 11 YouTube categories. Each heatmap represents a different category, with color intensity indicating $log_{10}$(Views).

Our analysis reveals several consistent patterns across categories. The maximum number of views ranges from 182 million to 1.48 billion views. The average number of views for most categories

falls between 63,000 and 251,000 views. Interestingly, for almost all categories, maximum views occur at very short sequence lengths (mostly 1) and low number of countries (2), suggesting that brief, targeted content can achieve high viewership.

Each category exhibits unique characteristics. The Games category demonstrates the highest maximum views (about 1.48 billion views) and one of the highest average views, indicating high engagement. In contrast, the News category contains the largest number of videos but shows a lower average view count, suggesting a high volume of content with more moderate individual performance. The Music category, despite having the fewest videos, maintains a competitive average view count, indicating that music videos tend to perform well relative to their number. The NonCategory exhibits the lowest average views, which might be expected for content that doesn't fit into standard categories.

The relationship between sequence length, country reach, and views varies across categories. Across most categories, shorter sequence lengths (0-20) tend to have higher average view counts. The Games category shows particularly high performance for short sequences (about 31,000 views on average). Regarding country reach, videos reaching 6-10 countries often have the highest average views across categories. There's a consistent decline in average views as the number of countries increases beyond 15, suggesting that very broad international appeal is rare.

**Studying Popularity and Reach**   We looked at how different video categories do across popularity and international reach using a grid of pie charts (Figure A7). Each pie chart represents a different quantile combination of video popularity (views) and international reach (number of countries), providing a comprehensive view of category distribution across various levels of success.

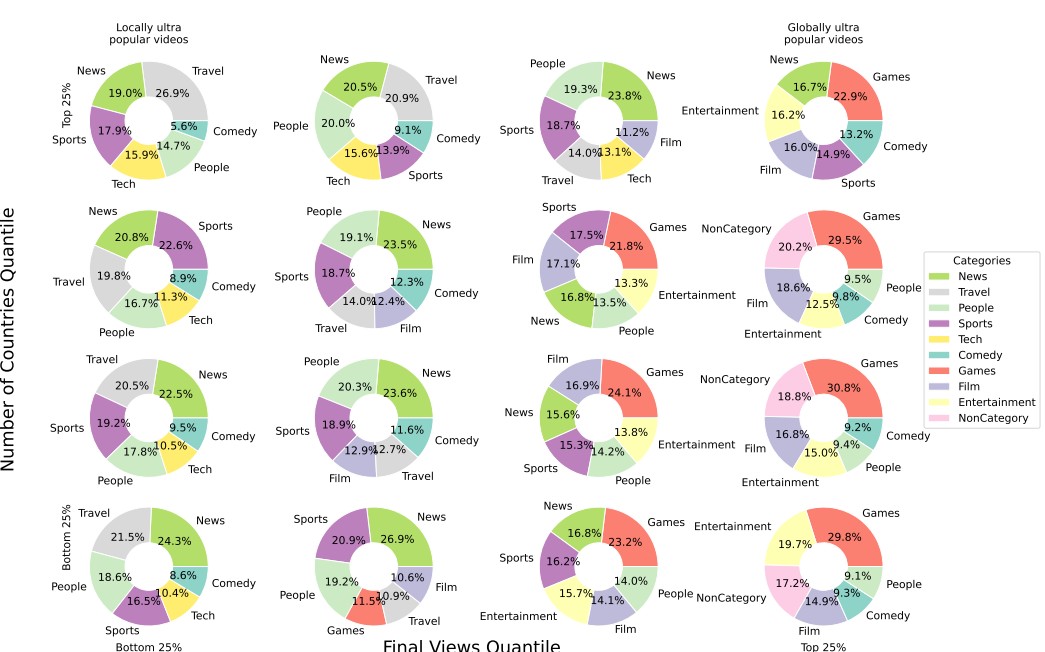

Figure A7: 4x4 grid of pie charts showing the distribution of video categories across different quantiles of popularity (views) and international reach (number of countries).

Our analysis uncovers the universal appeal of certain video categories across different levels of popularity and geographical spread. Notably, "People," "News," and "Sports" stand out, appearing in the majority of quantiles, indicating their widespread popularity. "Comedy" and "Film" also show strong presence, suggesting their content resonates across various levels of success and international reach.

A.6   ABLATION STUDY

A detailed ablation study was conducted to examine the effects of key model hyperparameters on prediction accuracy. This investigation focused on the interaction between the embedding type used

for near example retrieval, the count of these examples, and the temperature parameter of the Large Language Model (LLM). By methodically adjusting these parameters, the study aimed to reveal how variations in these elements influence the framework's predictive capabilities. The analysis closely examines the impact of embedding types, whether sourced from video descriptions or titles, the strategic selection of near example quantities, and the temperature settings within the LLM, providing a thorough examination of their combined effects on performance. This study is crucial as it illuminates the framework's operational nuances and informs potential adjustments, enhancing its effectiveness in the intricate task of video content analysis and prediction. Through this rigorous analysis, we aim to explore the framework's responsiveness to different hyperparameters, contributing to a nuanced understanding of its predictive mechanisms. This endeavor is not about optimizing the model per se but about uncovering how the framework behaves under varied conditions, offering valuable insights into its structure and function.

### A.6.1 Impact of embedding type on model performance

In our study of our proposed framework, we focused on understanding how the choice of embedding type, whether video descriptions or titles, used to find similar videos affects the accuracy of predicting video popularity. It's important to note that while both methods use full video descriptions, the key difference lies in how these similar videos are identified. This study looks into how choosing between video descriptions or titles to find similar videos affects prediction accuracy, showing that using titles to retrieve examples, surprisingly make our pipeline's predictions better.

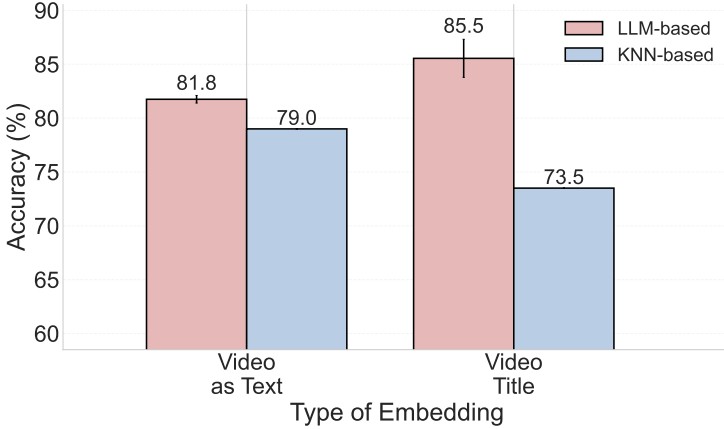

Figure A8: This graph compares how well our pipeling and near-examples models predict video popularity using 10 examples each, showing that using titles to find similar videos works better, even though both methods use examples containing full video descriptions as input. The only difference lies in how we find these examples: using titles or descriptions.

We report the findings in Figure A8, which show a nuanced resut of different kinds retrieved examples. Video-to-text achieved a mean accuracy of 81.75%, showcasing a consistent prediction capability with a standard deviation of 0.35%. This suggests that descriptions provide a reliable basis for similarity matching, albeit with a marginally lower accuracy compared to titles. Conversely, title embeddings yielded a higher mean accuracy of 85.5%, indicating their effectiveness in accurately identifying highly popular videos, albeit with increased variability, as evidenced by a standard deviation of 1.77%. This discrepancy may stem from the complexity and length of generated video descriptions, which could introduce extraneous information, diluting the core elements necessary for precise similarity matching. Titles, being more succinct, appear to offer a more focused approach for example retrieval, likely due to their ability to encapsulate the video's essence more directly. In contrast, the KNN model exhibited a mean accuracy of 79% with video descriptions and 73.5% with titles, highlighting a different pattern of performance that underscores the importance of model choice in leveraging embedding types effectively. This means that the way we select similar videos is key, and using titles, which are shorter, might predict popularity better by focusing on the main points of the video. The study shows that how we choose similar videos can make a big difference in how well our

pipeline works, suggesting that we should think carefully about how we find these videos to improve predictions.

### A.6.2 IMPACT OF NUMBER OF NEAR EXAMPLES ON MODEL PERFORMANCE

In this part of our study, we looked at how changing the number of similar videos (near examples) used by the our pipeline affects its ability to predict video popularity. Near examples are similar videos retrieved from the database that the model uses to identify patterns and make predictions. This analysis aims to determine the relationship of near examples, prediction accuracy and computational efficiency. We kept everything else the same and only changed the number of these examples to see how it impacts accuracy and how efficiently the model works.

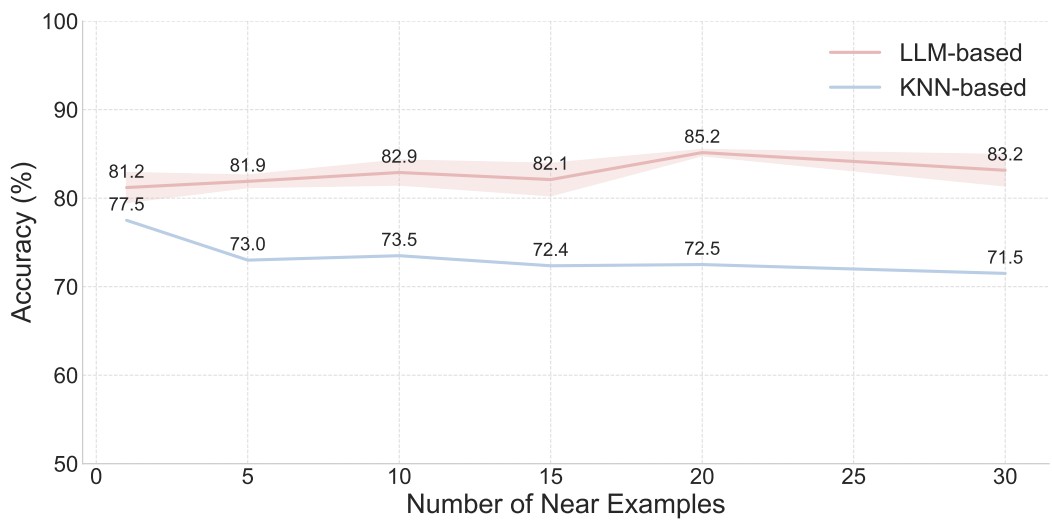

Figure A9: Impact of the number of near examples on accuracy. The plot shows mean accuracy and standard deviation for 1, 5, 10, 15, 20, and 30 near examples.

We tested using different numbers of near-examples, from 1 to 30, and found that more examples can actually help the model predict better, even though they might not seem as accurate when used in simpler models. Figure A9 shows this. The model's accuracy changes as we use more examples, with the best accuracy at 20 examples, suggesting this number might be just right for our model.

Interestingly, as we add more examples, the model's predictions become a bit less consistent, but it gets better at predicting overall. This means that even if more examples don't always lead to better results in simpler models, the LLM can use them to understand videos better and make more accurate predictions. However, using too many examples can actually make predictions a bit worse, showing there's a sweet spot at 20 examples where the model is both accurate and consistent. This finding is important because it shows that the LLM can use more information to improve, but there's a point where adding more doesn't help as much. It also reminds us that while more examples can help, we need to consider how much work the model has to do. Looking at how different types of videos respond to more examples could help us fine-tune the model even more, making it better at predicting video popularity.

### A.6.3 IMPACT OF TEMPERATURE ON MODEL PERFORMANCE

In language models, temperature is a hyperparameter that controls the randomness of the model's output. A lower temperature makes the model more deterministic in its predictions, while a higher temperature increases randomness and creativity. In the context of LLM based video popularity prediction, temperature plays a crucial role in balancing between making consistent, safe predictions and exploring more diverse, potentially insightful outcomes. Finding the optimal temperature is essential for maximizing the model's predictive accuracy while maintaining its ability to generalize across various video types and popularity patterns.

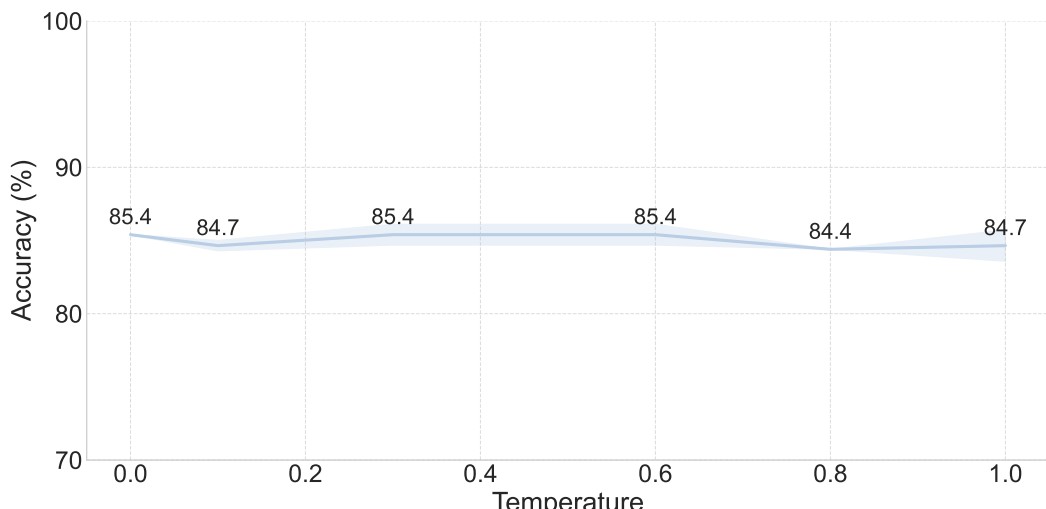

Figure A10: Impact of temperature settings. The plot shows mean accuracy for temperatures ranging from 0.0 to 1.0.

We looked at how different temperature settings affect the model's accuracy to find the best balance. Figure A10 shows that the model works well across a range of temperatures, with 0.3 and 0.6 being optimal, both hitting 85.4% accuracy with little variation. This means the model can handle different temperatures well, but there's a slight dip at 0.8 that needs more study. Future work could look closer at temperatures between 0.6 and 1.0 to understand why and improve predictions.

## A.7 ADDITIONAL ABLATION STUDY - SUPERVISED MODEL

This section presents a detailed ablation study of our supervised model for video popularity prediction. By systematically analyzing the performance of different feature combinations, we aim to identify the most effective features and understand the trade-offs between model complexity and prediction accuracy. To investigate the effectiveness of different feature combinations in predicting video popularity, we conducted a series of experiments using various feature sets. We analyzed the performance of individual features, pairwise combinations, triple combinations, and complex feature sets. This comprehensive analysis aims to identify the most effective feature combinations and understand the trade-offs between model complexity and prediction accuracy.

## A.8 ADDITIONAL ABLATION STUDY - BASELINE MULTIMODAL MODEL

| Feature | Embedding model | Embedding Size |
| --- | --- | --- |
| Text (Title, Description, Captions) | MPNet | 768 |
| Image (Thumbnail) | CLIP | 512 |
| Video (Key Frame Aggregation) | VideoCLIP | 512 |

Table A1: Baseline Embeddings

To establish a strong foundation for comparison, we implement a baseline multimodal model that leverages deep learning techniques to predict video popularity. The architecture of this baseline model is described in Table A2. The model consists of several key components, including a preprocessing layer to handle variable input sizes, main layers to learn complex representations, dimension matching layers to facilitate residual connections, an attention layer to weigh the importance of different parts of the input data, and a final classification layer to produce the output.

| Component | Details |
|---|---|
| **Preprocessing Layer** | Uses a linear transformation to map input data to a fixed dimensionality (processed_dim). |
| **Main Layers** | Composed of fully connected layers with batch normalization and ReLU activation. Includes dropout for regularization. Layers are sequentially connected with increasing reduction in dimensionality ($1024 \rightarrow 512 \rightarrow 256 \rightarrow 128$). |
| **Dimension Matching Layers** | Linear layers that adjust dimensions to enable addition of residual connections at each main layer stage. |
| **Attention Layer** | Consists of a linear transformation, a tanh activation, and a softmax output to produce attention weights. |
| **Final Classification Layer** | A fully connected layer that takes the attended features and outputs the final classification results. |
| **Overall Model Architecture** | Input data is processed through layers that include preprocessing, main processing with residuals, attention application, and final classification. |

Table A2: Baseline Multimodal Model Description

The baseline model utilizes various embeddings to represent the different features of the video data, as detailed in Table A.8. For textual features, such as the title, description, and captions, the MPNet model is employed to generate embeddings of size 768. Visual features, including the thumbnail, are processed using the CLIP model, resulting in embeddings of size 512. Finally, the video content is represented using key frame aggregation and the VideoCLIP model, producing embeddings of size 512. The baseline multimodal model serves as a robust point of comparison for our proposed framework, allowing us to assess the performance improvements achieved through the integration of VLMs and LLMs in the video popularity prediction task.

### A.8.1 INDIVIDUAL FEATURE PERFORMANCE

We begin by examining the predictive power of each feature type in isolation. Table A3 presents the performance scores for individual features.

| Feature | Score |
|---|---|
| Thumbnail | 0.77 |
| Video | **0.79** |
| Title | 0.75 |
| Description | **0.79** |
| Caption | 0.76 |

Table A3: Performance Scores for Individual Features

As shown in Table A3, video features and description features achieved the highest individual performance with a score of 0.79, followed closely by thumbnail features (0.77) and caption features (0.76). Title features showed the lowest individual performance at 0.75, suggesting that while titles contribute to prediction, they may not be as informative as other features when used alone.

### A.8.2 FEATURE COMBINATION PERFORMANCE

Next, we explore the synergistic effects of combining different feature types. Table A4 illustrates the performance scores for various feature combinations.

The combination of title features with other modalities consistently improved performance. Notably, the combination of thumbnail, description, and caption features achieved the highest score of 78%, demonstrating the complementary nature of these modalities in predicting video popularity.

| Combination | ⟨TITLE⟩ ⟨≡⟩ | ▷ | 🖼 | ❝≡❞ |
|---|---|---|---|---|
| - | **75** | **75** | 73 | 74 |
| ⟨TITLE⟩ ⟨≡⟩ | - | **76** | **76** | 72 |
| ⟨TITLE⟩ ⟨≡⟩ & ▷ | - | - | **77** | 76 |
| 🖼 & ▷ & ⟨TITLE⟩ ⟨≡⟩ | - | - | - | **78** |

Table A4: Performance analysis of multimodal feature combinations for video popularity prediction. Icons represent title and description (⟨TITLE⟩⟨≡⟩), video content (▷), thumbnail (🖼), and caption (❝≡❞). Bold numbers indicate the highest score in each row.

### A.8.3 COMPLEX FEATURE COMBINATIONS

Finally, we investigated the impact of combining four or five feature types. Table A5 shows the performance scores for these complex feature combinations.

| Feature 1 | Feature 2 | Feature 3 | Feature 4 | Feature 5 | Score |
|---|---|---|---|---|---|
| Videotext | Thumbnail | Video | Title | Description | 0.81 |
| Videotext | Thumbnail | Video | Title | Caption | 0.81 |
| **Videotext** | **Thumbnail** | **Video** | **Description** | **Caption** | **0.83** |
| Videotext | Thumbnail | Title | Description | Caption | 0.82 |
| Videotext | Video | Title | Description | Caption | 0.80 |
| Thumbnail | Video | Title | Description | Caption | 0.82 |

Table A5: Performance Scores for Complex Feature Combinations

The combination of videotext, thumbnail, video, description, and caption features achieved the highest score of 0.83. However, it's important to note that this score is not significantly higher than some of the triple feature combinations, suggesting a point of diminishing returns in terms of prediction accuracy as we increase feature complexity.

These results highlight the importance of considering multiple modalities in video popularity prediction. While individual features provide valuable information, the combination of complementary features leads to improved prediction accuracy. However, the experiments also reveal a trade-off between model complexity and performance gains, as the most complex feature combinations do not necessarily yield significantly better results than some simpler combinations. The analysis suggests that careful feature selection and combination can lead to efficient and effective video popularity prediction models. This combination likely captures a diverse range of information about the video content, including visual appeal, textual context, and spoken content.

These findings suggest that while incorporating multiple modalities can improve prediction accuracy, there is a point of diminishing returns. Future model development should focus on optimizing the balance between feature complexity and performance gains, potentially prioritizing the most informative feature combinations identified in this study.

### A.9 COMPARISON WITH VISION-LANGUAGE LARGE MODEL

To further strengthen our contribution, we conducted additional experiments using an advanced Vision-Language Large Model (VLLM), Gemini. Specifically, we used the Gemini 1.5 Pro model, accessed as API in Vertex library. The input to the model included a combination of the summariser prompt and the final prediction prompt. The sequential frames were aligned with the video's existing captions to ensure that visual and verbal elements were synchronized before being fed into the LLM

to generate the video-to-frame summary for the entire video. This process closely follows the steps described in Section 3.2 where "Frame Extraction" step (extracting 5 frames per minute) and the "Caption Matching and Data Integration" step were employed to align captions and frames for LLM processing. Subsequently, "Frames to Text Conversion and Summarization" step generated the final video summary using an LLM call. For prediction, we used the same final prompt detailed in Figure A2, ensuring consistency with our primary method.

These experiments confirmed that our strategy—incorporating sequential prompting, hypothesis generation, and supervised signals—consistently improves prediction performance, even with this state-of-the-art model (see Figure A11. This underscores the generalizability and robustness of our approach across different model architectures.

Notably, this result demonstrates that more advanced models do not inherently outperform others across all aspects; rather, each model tends to excel in specific areas. This highlights the importance of strategically combining models based on their unique strengths to achieve optimal results. Additionally, the prompting strategies developed in this work offer practical guidance for designing effective multimodal solutions. These insights not only inform future research on multimodal learning but also emphasize the value of integrating pre-trained models tailored to specific task requirements.

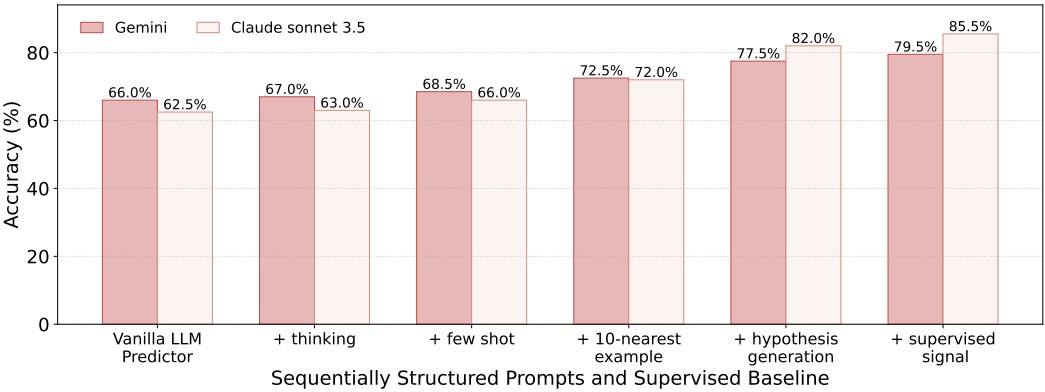

Figure A11: Performance evaluation of the Gemini 1.5 Pro model for video popularity prediction, demonstrating the impact of sequential prompting, hypothesis generation, and supervised signals.

## A.10    ADDRESSING LANGUAGE IMBALANCE IN TARGET CLASSES

To address concerns about potential language imbalance confounding our classification of 'local hit' and 'global big hit,' we provide a detailed analysis of the dataset's language distribution and its potential impact on model performance.

Figure A12 shows the distribution of target classes across major languages in the entire dataset and the subset used in our experiment. English constitutes approximately 40% of the 'global big hit' category and only 15% of the 'local hit' category. This distribution suggests that while English is overrepresented in the 'global hit' category compared to other languages, the difference is not as extreme as initially assumed. Moreover, non-English languages contribute substantially to both categories, reflecting a reasonably balanced dataset in terms of language diversity.

We also analyzed prediction accuracy by language to evaluate whether the model's performance is significantly influenced by the representation of each language in the dataset. Figure A13 presents the prediction accuracy for both 'local hit' and 'global big hit' categories across a variety of languages. The results show that the model achieves comparable accuracy for both major (e.g., English, Spanish) and minor (e.g., Swedish, Indonesian) languages, with no significant bias toward English or any other dominant language. These findings suggest that language representation does not disproportionately affect predictions.

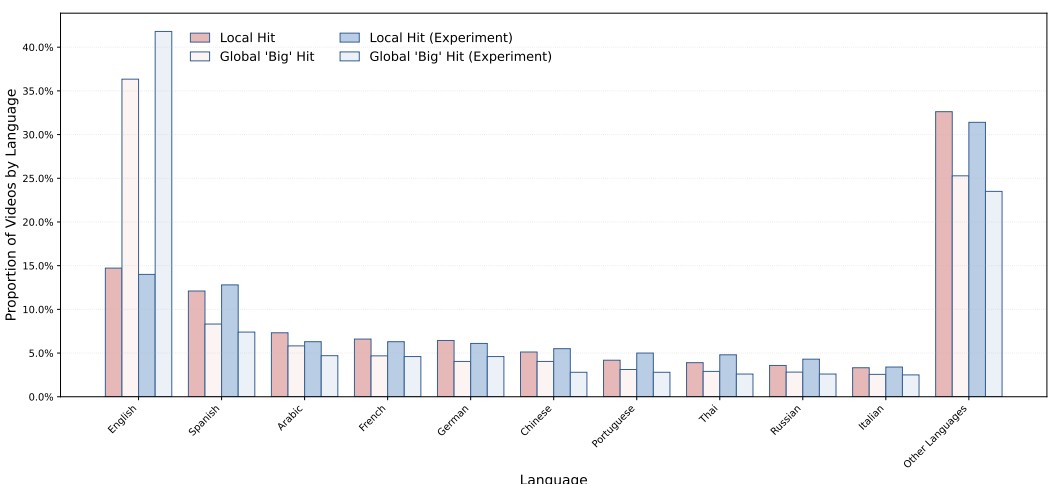

Figure A12: Distribution of 'local hit' and 'global big hit' categories across major languages. English represents 40% of the 'global big hit' category and 15% of the 'local hit' category, while other languages (e.g., Spanish, Hindi, French) also contribute significantly to both categories.

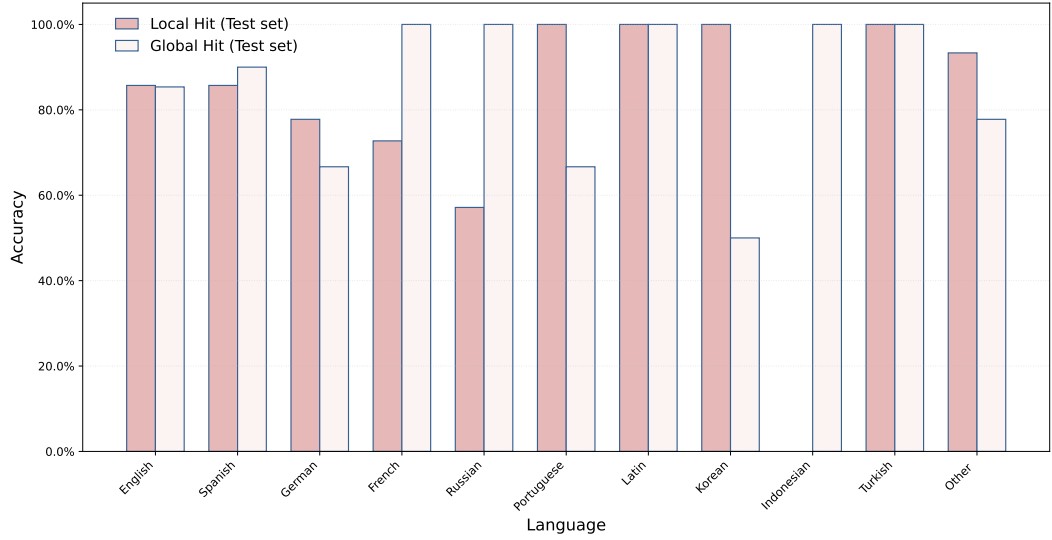

Figure A13: Prediction accuracy for 'local hit' and 'global big hit' categories across languages. The results show comparable model performance across major (e.g., English, Spanish) and minor (e.g., Swedish, Indonesian) languages, indicating that prediction accuracy is not significantly influenced by language representation.

