# OpenReview forum: "Large Language Models Are Natural Video Popularity Predictors"
_ICLR.cc/2025/Conference — Submitted to ICLR 2025_

### Official Review · Reviewer_1Pq3 · 2024-11-02

**Soundness:** 2
**Presentation:** 1
**Contribution:** 3
**Rating:** 3
**Confidence:** 4

**Summary:**

Previous approaches aiming at video popularity prediction have only considered engagement intensity, that is view count. However, due to cultural, social, and temporal factors, a video that is popular in one region may not be popular in another. Taking account of this, this paper proposes considering both view count and the number of countries a video has trended in (geographic spread) for predicting video popularity. With a well-designed sequential llm prompting method and hints from a supervised model, the proposed method predicts video popularity taking as input only the intrinsic video content, including title, description, caption and video frames. The empirical results show that how the proposed method improves over a supervised baseline model on a newly collected dataset, Video Popularity Dataset (VPD).

**Strengths:**

- Considering both engagement intensity and geographic spread makes sense.
- The authors collected a valuable dataset, VPD. This would foster future work.
- The empirical results explicitly show the improvements from the hybrid approach using both llm prompting and supervised signals for video popularity prediction.

**Weaknesses:**

This paper has the following issues:

1. The authors did not compare the proposed method with any previous approaches.
2. The main contribution of this paper is its consideration of geographic spread - how many countries a video has trended in - as well as engagement intensity, that is view count, in measuring video popularity. However, they seem to be quite orthogonal to me, but the authors used only a single score integrating both two factors. I am not sure why they used separate scores measuring each. Especially, While the authors mentioned in section A.3 that there is a positive correlation between them, but according to Figure A2, there are also non-negligible number of videos categorized into "locally ultra popular" and "globally moderately popular".
3. The writing needs to be improved. There are many redundant sentences aimed at increasing the length of the paper. For example, Section 5 contains too redundant statements about the hybrid approach using both llm prompting and supervised signals. Furthermore, there are too many cases where figures or tables contradict the text. To name a few, in Section A.3.2 (L803), the authors mentioned that the upper-right quadrant in Figure A3 is less populated than the lower-left quadrant, which contradicts the fact that there are 6,279 local hits videos and 7,360 global bit hits videos. There are also no 0.83 in Table A4 (L1187); the highest score is 0.78. The authors should also fix typos, e.g., please fix "red lines" in the Figure A3 caption to "blue lines". Overall, it feels as though the authors have submitted an unfinished draft that hasn’t been proofread.
4. There are too many missing details
- How did the authors differentiate the videos of classes 1 and 2 in the local hit cluster and the videos of classes 3 and 4 in the global big hit cluster? There are two defining factors, geographical spread and view count, and I am not sure about how they are integrated into a single score.
- The authors did not mention the training recipe and data for the supervised baseline model.
- L299: Please elaborate on the hypothesis generation function Phi_hypothesis, especially regarding the reinforcement learning technique used for it, if any (Section 5). And how many hypotheses are used for prediction?
- Please provide the exact final prompt text.
- How did the authors screen participants in the surveys?
- And please let me know exactly what questions were asked to the participants in the survey to score each metric.
- L508 "This feature enhances not only the performance but also the explainability of the model": This statement is not explicitly validated in the paper. The authors used only accuracy and precision/recall for evaluation.

**Questions:**

Please address the above concerns and I have the following suggestions:

1. It would be good to explicitly mention in the abstract that the model predicts geographic spread as well as view count.
2. It would be great to ablate the model design choice for each component. One of them would be replacing the MPNet base v2 model with other recent models to generate embeddings for retrieving near examples (L289-290).
3. Figure 1 does show the entangled improvement from both 10-nearest examples and hypothesis generation. Even though Figure A8 compares the results of 1-nearest example and 10-nearest examples, it would be great to add the result with 10-nearest example only also to Figure 1.

One minor question:
L185-L186: I am not sure whether sampling an equal number of videos from each class is appropriate for testing; it does not reflect the real-world video distribution.

**Details Of Ethics Concerns:**

The paper did not explicitly discuss about whether the collected dataset, VPD, has harmful video content or how to filter out the harmful and non-ethical video content.

---

> ### Author Response · Authors · 2024-11-25
>
> Thank you for your careful evaluation of our work and for highlighting its strengths and areas for improvement. We are grateful for the opportunity to clarify and refine the paper in response to your thoughtful feedback. Please find our point-by-point response below.
>
> ### **Strengths**
>
> We sincerely thank the reviewer for acknowledging the strengths of our work. We are glad that the integration of engagement intensity and geographic spread as dimensions of popularity resonated with you, as it represents an essential conceptual innovation. Your recognition of the value of the Video Popularity Dataset (VPD) as a resource for future research is greatly appreciated. Additionally, we are pleased that the empirical results demonstrating improvements from our hybrid approach—leveraging both LLM prompting and supervised signals—were found to be a meaningful contribution. To further enhance these strengths, we have included additional details about our dataset and clarified the improvements achieved through our approach in the revised manuscript. Your positive feedback inspires us to refine and strengthen these aspects of our work further.
>
> ### **Weaknesses**
>
> We sincerely thank the reviewer for highlighting these areas for improvement, which provide us with an opportunity to address key gaps and enhance the quality and rigor of our work. Please find the clarifications and details of our revisions in the following point-by-point responses.
>
> #### **1. Lack of comparison with previous approaches:**
>
> We appreciate the reviewer's comment and understand the importance of benchmarking our method against existing approaches. While direct comparisons with prior works are challenging due to the uniqueness of our dataset and task (i.e., distinguishing global big hit vs. local hit videos), we acknowledge that our hybrid framework builds upon ideas and strategies from previous research. Specifically, our work combines prompting techniques, supervised signals, and KNN-based examples in a novel way to tackle the complexity of the task. This integration is what sets our approach apart and contributes to its performance improvements.
>
> Additionally, while our focus was not on direct comparisons with specific models, the supervised baseline used in this study serves as a meaningful benchmark, particularly since video popularity prediction has traditionally been approached as a supervised learning problem. By demonstrating that our hybrid approach outperforms this strong baseline while also generating reasonable explanations (i.e., hypotheses), we provide compelling evidence of its effectiveness.
>
> To address your concerns more thoroughly, we will expand the discussion in the revised manuscript to highlight how our ablation studies align with strategies proposed in prior works and how our approach leverages these strategies to achieve state-of-the-art results. We will also clarify that the supervised model we used can be viewed as a benchmark for future studies and provide a more detailed comparison of its performance relative to our hybrid framework. Thank you for bringing this point to our attention, and we hope these additions will strengthen the manuscript.
>
> #### **2. Integration of engagement intensity and geographic spread into a single score:**
>
> We appreciate the reviewer’s observation regarding the use of a single score for integrating geographic spread and engagement intensity and the resulting focus on two primary classes, “global big hit” and “local hit.” Ideally, we would evaluate all four quadrants (e.g., including “local big hit” and “global hit” videos additionally) to explore the complete spectrum of popularity. This would then clearly reflect both dimensions. However, since our main contribution lies in the development of a robust pipeline integrating collective prompting strategies, including novel approaches such as incorporating supervised signals and hypothesis generation, we simplified the experimental design to a binary classification task focusing on the two most distinctive categories. We hope this clarification addresses your concerns.
>
> That said, given the robustness of our results, including consistent performance when tested with advanced models like Gemini (see our responses to Weakness 5 from Reviewer 1 and Weakness 2 from Reviewer 2), and the high quality of hypotheses and reliability against hallucination issues as verified by human evaluators, we are confident that our pipeline would be comparable to or outperform traditional supervised approaches even in a multi-class prediction setting involving all four quadrants. If the reviewer believes that extending the analysis to include multi-class prediction would significantly enhance the value of the paper and improve its suitability for acceptance at the conference, we would be more than willing to conduct this analysis and include the results in the camera-ready version. Thank you for identifying this potential area for improvement.

---

> ### Author Response · Authors · 2024-11-25
>
> #### **3. Writing and inconsistencies:**
>
> We appreciate your detailed feedback regarding the writing and presentation of the manuscript. We will revise the paper to eliminate redundancy, particularly in Section 5, and ensure all figures, tables, and references align accurately with the text. Specifically:
>
> + We will address the discrepancy in Section A.3.2 concerning the distribution of local and global hits.
>
> + The reported scores in Table A4 will be carefully reviewed and corrected as needed.
>
> + Errors such as the reference to "red lines" in the caption of Figure A3 will be corrected.
>
> + A thorough proofreading of the entire manuscript will be conducted to enhance clarity, conciseness, and consistency.
>
> Thank you for highlighting these areas for improvement, which will help us significantly enhance the overall quality of the paper.
>
> #### **4. Missing details:**
>
> We appreciate your request for more detailed explanations. Here’s how we plan to address these points:
>
> #### *4-1. Differentiation of classes 1, 2, 3, and 4:*
>
> We appreciate the reviewer’s insightful observation regarding the differentiation of classes and the justification for integrating intensity and spread into a single prediction scale. The categories "locally ultra-popular" and "globally moderately popular" indeed represent important real-world phenomena, and our intention was to construct a prediction task that captures the complexity and challenge of these dimensions. However, for the purposes of simplicity and interpretability, we devised a binary classification task that focuses on two highly distinctive quadrants: globally highly popular videos and locally popular videos.
>
> To create these groupings, we used relative thresholds by binning videos based on their reach (i.e., the number of countries where a video was popular) and view counts. Each dimension was divided into quartiles, resulting in four groupings: Q1-Q1, Q1-Q4, Q4-Q1, and Q4-Q4. This approach—using medians or quartiles to discretize continuous variables—is widely used and ensures a balanced and robust categorization of videos for our analysis. The binary classification task was then derived by selecting the most contrasting groups, namely Q4-Q4 (globally highly popular) and Q1-Q1 (locally popular).
>
> While we acknowledge the potential benefit of a more explicit two-dimensional construct, we believe this simplification aligns with our goal of demonstrating the robustness and effectiveness of our pipeline in distinguishing videos with the most disparate characteristics. We also believe this binary framework emphasizes the practical utility of our method by focusing on the key distinctions that matter most for understanding video popularity at scale.
>
> If you think this explanation is clear and reasonable, we will clarify these details, including how we binned videos into categories and devised the one-to-four unidimensional scale for prediction, in the revised manuscript. Thank you for your sharp and constructive feedback, which helps us refine the presentation of our methodological approach.
>
> #### *4-2. Training details for the supervised baseline:*
>
> Thank you for highlighting the need to improve the details provided about our baseline approach. While we initially included the exact design and dimensions of the model in Table A2 (Baseline Multimodal Model Description) and Section A.8, we acknowledge that key training hyperparameters, such as the learning rate, optimizer, and early stopping, were not explicitly mentioned. To address this, we have updated Section 4.1 under Implementation Details to include the following information: *"For the supervised baseline model training, we used a learning rate of 0.001 with the Adam Optimizer and implemented early stopping with the patience of six epochs. See Section A.8 for more details."* These additions provide a clearer understanding of our training process and enhance the reproducibility of our work. We believe these updates significantly improve the clarity and rigor of the manuscript, and we appreciate your suggestion to include these details.

---

> ### Author Response · Authors · 2024-11-25
>
> #### *4-3. Hypothesis generation function:*
>
> Thank you for highlighting the need to clarify the hypothesis generation process. In our paper, we define hypothesis generation as the process of prompting the LLM to create a set of hypotheses, $\mathcal{H} = \{h_j\}_{j=1}^M$
>
> based on the near examples $\mathcal{E}_\text{near}$
>
> using a hypothesis generation function $\Phi_\text{hypothesis}$. In essence, the hypothesis generation function is the specific prompt we design and use in conjunction with the LLM to produce these hypotheses. These hypotheses represent potential signals or factors associated with video popularity, both for popular and non-popular videos.
>
> In our experiments, we explicitly instructed the model to generate 4 hypotheses for each case. The hypothesis generation prompt, $\Phi_\text{hypothesis}$, was designed as follows: *“Create four hypotheses about why videos in the <similar_examples> group are categorized as ultra-popular (Evaluation=4) and why some remain in the basic popularity category (Evaluation=1). Identify patterns from these example videos, generalize the factors that contribute to high popularity, and explain why some videos achieve only basic popularity.”* This structured approach ensures consistency and allows us to systematically analyze the generated hypotheses in the context of video popularity prediction.
>
> Regarding reinforcement learning, while we did not explicitly implement reinforcement learning (RL) in this work, we recognize that the iterative nature of refining hypotheses based on their correctness and impact resonates with RL techniques. In Section 5, we mentioned: *"While our model already shows substantial improvements, further refinement of the hypothesis generation process could enhance accuracy, particularly by incorporating more advanced reinforcement learning techniques."* To bring more clarity, we have added the following explanation: *"More specifically, the LLM acts as an agent generating hypotheses about video popularity factors, where each hypothesis represents an action within the state space of possible predictions. The prediction accuracy serves as a reward signal, guiding the system to learn which types of hypotheses are most effective."*
>
> We hope these clarifications address your concerns. Please let us know if you have any additional suggestions or concerns!
>
> #### *4-4. Final prompt text:*
>
> The exact text of our final prompt is now included in Figure A2. See https://drive.google.com/file/d/1nOyjd4BWwftPnNyKFjTYPYeIkzpBX3u2 for your convenience!
>
> #### *4-5. Survey screening and questions:*
>
> Thank you for your inquiry regarding our survey procedure. We targeted participants with at least a Master's degree based in the United States, without implementing additional pre-screening criteria for participant selection. After participants completed their evaluations, we assessed their attentiveness to the video's details through a series of post-evaluation screening questions. These questions included: (1) Identifying the correct title of the video; (2) Recognizing the activity depicted in the video; and (3) Recalling the specific evaluation metric they were asked to assess. Participants who answered all three questions correctly were classified as having "passed" the screening. Importantly, the results were nearly identical regardless of whether participants passed the screening, underscoring the robustness and reliability of our approach.
>
> To enhance transparency and facilitate reproducibility, we have included all the survey questions in the appendix. Additionally, we will share the full Qualtrics survey to assist others who may wish to use our experiment as a template. For your convenience, an example questionnaire is provided here: https://drive.google.com/file/d/1dGZ3jX78450A9mw7uHlpAWzdhQu3A9u7.
>
> #### *4-6. Explainability validation (L508):*
>
> While we did not include quantitative metrics for explainability, we validated this feature through human evaluations of the generated hypotheses. We have updated the manuscript to clarify how these evaluations support our claims regarding interpretability (see lines 536-538).

---

> ### Author Response · Authors · 2024-11-25
>
> #### **Additional Suggestions**
>
> + **Explicit mention of predicting geographic spread and view count in the abstract:** Thank you for this suggestion. We agree that explicitly stating the model’s ability to predict both geographic spread and view count will clarify the scope and contribution of our work. We have added this point in the abstract in the revised version.
>
> + **Ablating the model design choice for each component:** Thank you for this insightful suggestion. We agree that exploring alternative embeddings, such as replacing the MPNet base v2 model with other recent models, would enhance the comprehensiveness of our ablation studies. While we are unable to incorporate this additional analysis in the current submission, we commit to conducting it for the camera-ready version if the paper is accepted. This will provide a richer understanding of how different embeddings impact performance.
>
> + **Adding results with 10-nearest examples to Figure 1:** Thank you for this suggestion! We have updated Figure 1 to include the results for 10-nearest examples along with 1-nearest example, as shown below. We hope this revised figure addresses your concern and makes the improvements from each component clearer.
>
> + **One minor question (L185-L186):** Thank you for raising this point about sampling. We recognize that sampling an equal number of videos from each class may not reflect real-world video distribution. However, this design choice was made to ensure balanced training and evaluation across classes, allowing us to focus on the nuanced distinctions between globally popular and locally popular videos without introducing potential biases from class imbalances. We believe this approach provides valuable insights into the model's core capabilities and ensures fair assessment across all conditions. We have clarified this rationale in the revised manuscript (see lines 199-201).

---

> ### Comment · Reviewer_1Pq3 · 2024-11-26
>
> Thank you for your thorough response, which significantly enhances the quality of the paper. However, I still have the following points that I would like you to answer:
>
> - While I acknowledge that the current experimental results demonstrate the robustness of the proposed pipeline, simplifying the task by predicting a one-dimensional score such as local hit and global big hit remains problematic. Especially, while you claim that the main contribution lies in considering both geographic spread and view count for video popularity prediction, the model does not need to consider both factors to solve the given binary classification problem; you can get the correct answer by predicting either the view count or the geographic spread.
> - Regarding the differentiation of classes 1, 2, 3, and 4, you did not address how the distinction between classes 1 and 2, and classes 3 and 4, are made.
> - Although you have added explanations regarding data collection, please clarify what efforts you made to filter out harmful video content. If you did not make any efforts, please explain the rationale behind this decision.

---

> > ### Author Response · Authors · 2024-11-27
> >
> > Q1) Potential issue on the one-dimensional score not capturing both dimensions:
> >
> > > Thank you for highlighting this important point. Your concern is valid, and we acknowledge the need to provide further clarification regarding the model's ability to account for both dimensions—view count and geographic spread—in its predictions. While we initially did not address this explicitly, our experiments consistently demonstrated that the model incorporates both factors. For example, consider a football game during the FIFA World Cup versus a rivalry game in a professional South American league. While both videos might achieve similar view counts, the model correctly emphasizes the global appeal of the World Cup match while recognizing the localized attention of the league game.
> >
> > > Such behavior is also evident in our qualitative analysis (Section 4.3), where the model highlights not only attributes tied to engagement intensity (e.g., unique content features) but also aspects related to global reach (e.g., universal appeal and international recognition).
> >
> > > To make this more explicit, we have revised the entire section of qualitative analysis (Section 4.3), emphasizing how the model effectively integrates these dimensions for its predictions. We hope this addition provides clarity and assurance that the binary classification inherently requires and leverages insights from both geographic spread and view count.
> >
> > Q2) The differentiation of classes 1, 2, 3, and 4:
> >
> > > Thanks for another very sharp comment!
> >
> > > We initially designed the base prompt with only two classes. However, during experiments, we observed that the LLM set a very high standard for the Global 'Big' Hit class, leading to a noticeable bias towards the Local Hit class. Introducing buffer classes (1 and 2 for local hits and 3 and 4 for global hits) addressed this issue by: (1) allowing the model to express uncertainty through intermediate predictions, avoiding forced binary decisions; (2) creating a more granular classification system that better reflects real-world ambiguities between extreme categories; (3) enabling better-calibrated confidence levels by incorporating buffer zones between classes; and (4) establishing a smoother decision boundary between extremes, reducing the potential for overly rigid classifications. This adjustment led to improved prediction performance, and thus we incorporated the four-class structure into our experimentation pipeline. Details of the class setup can be found in Section A.4 of the Appendix, which includes the final prompt used in our experiments.
> >
> > > Since we realized that this may be helpful for others who may conduct a similar study, we added these details in the paper, too (see footnote 2 in the revised manuscript).
> >
> > Q3) Potentially harmful video content in the dataset:
> >
> > > Thank you for raising this concern regarding the potential inclusion of harmful content in our dataset. The dataset was collected directly from YouTube's trending lists, curated by the platform's algorithms to highlight popular videos gaining traction in specific countries. While the platform's intervention and collective social mechanisms generally minimize the harmful content to be placed at the top daily videos, some videos may still be perceived as controversial by certain audiences.
> >
> > > In terms of the filtering efforts, we did not actively filter out specific types of content during data collection to preserve the dataset’s integrity and representativeness, reflecting the real-world distribution of trending videos. Applying subjective filters could introduce bias and reduce the dataset’s utility for analyzing trends, behaviors, or platform dynamics.
> >
> > > Further Rationale:
> > > - Reflecting Reality: Harmful content is part of the digital landscape and offers insights into how such videos gain popularity or engagement. Excluding them could skew analyses, particularly for studies on societal impacts or content moderation.
> > > - Generalizability: Including the full spectrum of trending videos allows the dataset to support diverse research areas, such as misinformation detection or content moderation strategies, where harmful content may be particularly relevant (if any).
> > > - Passive and Predictive Focus: The dataset is a comprehensive "record" of daily top videos across countries, not promoting or disseminating content. Our predictive pipeline focuses on understanding trends, not reproducing or amplifying content.
> >
> > > Ethical Considerations:
> > > When sharing the dataset, we will (1) acknowledge the potential risks associated with harmful content and (2) encourage researchers to handle such data responsibly, adhering to ethical guidelines.
> >
> > > If you feel that further clarification on this rationale and approach is necessary, we are happy to include these points in the manuscript (potentially in the Appendix) and the data collection section. Thank you again for highlighting this critical aspect.

---

> > > ### Author Response · Authors · 2024-11-27
> > >
> > > A fully revised version of the paper has been uploaded and is ready for your review.

---

> > > > ### Comment · Reviewer_1Pq3 · 2024-11-28
> > > >
> > > > Thank you for the response. Please add the reason why you did not actively filter out harmful content in the draft. Regarding the first concern,
> > > > - We cannot generalize from a few examples where the model accounts for both factors.
> > > > - Since the measure itself is not well defined, we cannot confirm the main contribution you intended to deliver through Figure 1. Although introducing each of prompt components improves the score, we cannot say that it enables the model to better predict both geographic spread and view count.

---

> > > > > ### Author Response · Authors · 2024-11-28
> > > > >
> > > > > I understand your suggestion to demonstrate that the model predicts each dimension (geographic spread and view count) independently. However, we do not claim that the model independently predicts each dimension. Rather, we emphasize that the model is designed to integrate signals from both dimensions whenever relevant. Considering both dimensions is particularly important, especially given that the global reach dimension has been entirely ignored in previous research. The ability to consider both dimensions—independently or jointly—is a strength of our model. Some cases may prioritize one dimension over the other, others may require independent consideration of both, and certain scenarios necessitate a joint understanding. While a deeper exploration of this balance could be an avenue for future work, our primary focus is on the model's capacity to adaptively incorporate signals from both dimensions, enabling nuanced and realistic predictions, while generating sounding hypotheses.

---

### Official Review · Reviewer_2kt8 · 2024-11-03

**Soundness:** 2
**Presentation:** 3
**Contribution:** 2
**Rating:** 3
**Confidence:** 4

**Summary:**

This paper presents an approach to predicting video popularity using LLMs. The authors argue that traditional methods, which rely on meta-information and aggregated user engagement data, fail to capture the contextual nuances that influence video popularity. To address this, the paper introduces a method that transforms visual data into sequential text representations using VLMs, allowing LLMs to process multimodal content. The authors evaluate their approach on a new dataset of 17,000 videos and demonstrate that their LLM-based method outperforms supervised neural networks and provides interpretable hypotheses for its predictions.

**Strengths:**

1. The paper propose an innovative use of VLMs to transform visual data into text, enabling LLMs to capture rich contextual information from video content, which traditional methods often overlook.

2. The proposed method achieved higher accuracy rates compared to supervised neural networks, with an 82% accuracy without fine-tuning and an 85.5% accuracy when combined with a neural network, showcasing the potential of LLMs in this domain.

3. The paper not only focuses on prediction accuracy but also on the interpretability of the predictions. The LLM generates hypotheses that explain its predictions, adding a layer of transparency and trust in the model's output.

4. The introduction of the Video Popularity Dataset, which includes a large number of videos with detailed metadata, view counts, and global reach metrics, is a significant contribution to the field.

5.  By considering both engagement intensity and geographic spread, the paper offers a more comprehensive understanding of video popularity than previous studies that focused solely on view counts.

**Weaknesses:**

The following are comments that combine my suggestions and PC's feedback for my suggestions! Thank you, PC!

1, It would be great for the authors to elaborate on the broader implications and potential applications of their work beyond just video popularity prediction. For example, you could ask how the techniques developed here might generalize to other multimodal prediction tasks, or how the interpretable hypotheses generated by the LLM could be applied in other domains requiring explainable AI.

2, From machine learning perspective, It would be great for the authors to provide a more detailed comparison with existing multimodal fusion techniques or  elaborate on the specific innovations in your approach, such as how the VLM-to-LLM pipeline differs from traditional multimodal fusion methods, and what unique challenges they addressed in combining visual and textual data for this particular task. It would be also great to highlight the broader lessons or insights about multimodal fusion that can be drawn from their work.

**Questions:**

Please answer the question from the weakness part and I would be happy to discuss with the authors about the questions and change my score if necessary.

---

> ### Author Response · Authors · 2024-11-25
>
> We sincerely appreciate your valuable insights and constructive suggestions, which have significantly guided our efforts to enhance the quality and clarity of the manuscript. Please find our point-by-point response below.
>
> ### **Strengths**
>
> We sincerely appreciate the reviewer’s recognition of the strengths of our work, including the innovative application of Vision-Language Models (VLMs) and Large Language Models (LLMs) to transform visual data into text, enabling a deeper contextual understanding of complex tasks like video popularity prediction. We are particularly pleased that our dataset and framework's contributions—such as achieving superior accuracy compared to traditional methods and providing interpretable, hypothesis-driven predictions—were highlighted. To further emphasize these strengths, we have added comprehensive details about our Video Popularity Dataset, showcasing its richness and unique value in facilitating cross-cultural and temporal analyses of video trends (see our response to Strength 1 from Reviewer 1 for more details). Moreover, our integrated approach, which combines engagement intensity and geographic spread, offers a broader conceptual framework that we believe can serve as a foundation for future research across domains. Thank you for your thoughtful feedback, which has allowed us to refine and strengthen our contributions further.
>
> ### **Weaknesses**
>
> #### **1. Broader implications and potential applications:**
>
> We appreciate the suggestion to elaborate on the broader implications and potential applications of our work. Beyond video popularity prediction, the techniques developed in this study, such as the VLM-to-LLM pipeline and hypothesis generation, have significant potential for generalization to other multimodal prediction tasks. For example:
>
> + **Social Media Analysis:** Our framework could be applied to analyze trends, sentiments, or engagement on platforms where multimodal content (images, videos, and text) is prevalent.
>
> + **Healthcare:** The interpretability of hypotheses generated by LLMs could enhance transparency in medical imaging and diagnosis, where explainable AI is critical for trust and adoption.
>
> + **Education:** The approach could aid in creating explainable feedback for multimodal student performance assessments or content personalization.
>
> + **Broader Research in Computational Social Science:** In social sciences, there is a growing need for computational methods that provide nuanced explanations beyond simple coefficients in regression models. The high-quality hypothesis generation enabled and demonstrated by our framework can serve as a basis for theoretical exploration, offering possible explanations and potentially transforming how researchers approach sociological and cultural questions.
>
> By addressing both practical applications and theoretical advancements, we believe our work opens pathways for impactful future research. These broader implications are now emphasized in the discussion section, offering a more comprehensive perspective on the potential contributions of our study. We sincerely thank the reviewer for encouraging us to unpack this important aspect, which we believe has significantly enhanced the overall quality of our paper!

---

> > ### Author Response · Authors · 2024-11-25
> >
> > #### **2. Comparison with existing multimodal fusion techniques:**
> >
> > Thank you for highlighting the importance of a detailed comparison with existing multimodal fusion techniques. Our work focuses on demonstrating how existing pre-trained models can be effectively combined to achieve state-of-the-art performance in a complex prediction task. This is particularly important in the current landscape, where abundant pre-trained models with varying capabilities often lack systematic guidance for application to specific end tasks. We see our study as a stepping stone toward addressing this gap.
> >
> > Below, we outline the specific challenges we address and the contributions of our approach:
> >
> > + **Use of VLMs for LLM-based end tasks:** Unlike prior approaches, our method employs visual-text outputs from Vision-Language Models (VLMs) to enable LLMs to perform an end task, namely classification. This approach has broader implications, particularly for tasks like video analysis, which require integrating visual data with contextual information often absent in datasets or platform metadata. By distilling video content into textual representations, our framework allows LLMs to reason and make predictions in ways that transcend traditional multimodal methods.
> >
> > + **Hybrid approach with pre-trained models:** Instead of developing new architectures, we demonstrate how existing pre-trained models can be integrated to enhance both statistical pattern extraction and contextual reasoning. Importantly, our hybrid framework (a) employs supervised learning for initial feature extraction; (b) leverages LLMs for reasoning and contextual understanding; and (c) integrates these components using an innovative augmentation technique. This approach differs from traditional multimodal fusion methods that often rely on embedding concatenation or end-to-end training. By leveraging contextual prompting, our method achieves high performance with limited data, offering efficiency and scalability.
> >
> > + **Advantages of our approach:** Our framework achieves superior results compared to pure machine learning models, delivering high accuracy rates while generating interpretable reasoning for its predictions. Unlike traditional multimodal fusion techniques requiring extensive labeled datasets, our approach effectively handles complex video analysis tasks with minimal training data.
> >
> > To further substantiate our contributions, we conducted additional experiments using an advanced Vision-Language Large Model (VLLM), Gemini. These experiments confirmed that our strategy—including sequential prompting, hypothesis generation, and supervised signals—consistently enhances prediction performance across both Gemini and our initial setup with Claude Sonnet 3.5. The figure here, https://drive.google.com/file/d/1pZjUr5SHD9a0Y-nPczr6adwxvJWzfVSI (also included in the appendix), highlights these results, further validating the effectiveness of our approach across state-of-the-art models.
> >
> > + **Lessons for the community:** Our work highlights the significant potential of leveraging pre-trained models and intermediate textual representations to solve multimodal tasks effectively. It also underscores the value of contextual prompting in enabling LLMs to overcome data limitations and perform reasoning-based predictions. Importantly, our findings emphasize that more advanced models do not inherently outperform others in all aspects; instead, each model may excel in specific areas. This insight suggests that strategically combining models based on their unique strengths is crucial for achieving optimal results. Moreover, the prompting strategies demonstrated in our work provide a practical framework and guidance for designing effective multimodal solutions. These insights can inform future research on multimodal learning, particularly in the integration of pre-trained models for diverse end tasks, and highlight the importance of tailoring model combinations to task-specific requirements.
> >
> > To clarify these contributions, we have extensively updated the manuscript, particularly the related work and discussion sections. We hope these revisions effectively address your concerns and help you positively consider the significance of our work.

---

> > > ### Author Response · Authors · 2024-11-26
> > >
> > > A fully revised version of the paper has been uploaded and is ready for your review.

---

### Official Review · Reviewer_HVCp · 2024-11-04

**Soundness:** 2
**Presentation:** 2
**Contribution:** 2
**Rating:** 6
**Confidence:** 4

**Summary:**

1. The paper addresses a new video popularity prediction task of distinguishing between globally and locally popular content.
2. The authors develop a Video Popularity Dataset (VPD)  of 17,000 YouTube videos, with titles, descriptions, and detailed metadata, along with the countries the video was trending in amongst 109 countries.
3. The authors transform video content into textual representations using Vision-Language Models (VLMs) to bridge the modality gap between video data and LLMs.
4. The authors test extensively on prompting paradigms for the video popularity prediction task, including Thinking, ICL, Hypothesis generation, and supervised signal into the prompt.

**Strengths:**

1. The paper introduces a new dataset and task for local and global popularity prediction.
2. The insights like reach vs popularity, ablation studies, are useful.
3. The authors conduct human studies to verify video verbalization and hypothesis generation.

**Weaknesses:**

1. The frame-to-text method is not novel, there are multiple works [1,2,3,4] that use titles, channel, generated visual captions, ASR of youtube videos for tasks like Views, Likes/Views, replay predictions [2,3,4].
2. There is no explanation of the 17k videos and metadata collection, one subsection on the tools and data sources is necessary.
3. The dataset consists of only global and local hits, and not negative samples, looking at locally/globally popular videos from the examples (and https://kworb.net/youtube/trending_overall.html#google_vignette , since any other source was not mentioned in the paper) there seems to be a huge language bias (Hindi/Korean/.. languages being local hits)
4. The human evaluation is not signifcant for both tasks,14,11 subjects passed the screening and they were only given 2 video-caption pairs, totalling 28, 22 interactions only.
5. The technical contributions are severely limited, the authors only show combinations of known prompting strategies for evaluation.

[1] https://aclanthology.org/2023.emnlp-main.608/
[2] https://arxiv.org/abs/2309.00378
[3] https://arxiv.org/abs/2405.00942
[4] https://openreview.net/forum?id=TrKq4Wlwcz

**Questions:**

1. What steps were taken to filter the videos for this dataset? Languages, Tags
2. How was the dataset collected
See weaknesses

---

> ### Author Response · Authors · 2024-11-25
>
> Thank you for your detailed and thoughtful feedback on our work. Your comments have helped us identify key areas for improvement, and we appreciate the opportunity to address them comprehensively. Please find our point-by-point response below.
>
> ### **Strengths**
>
> #### **1. Introduction of a new dataset and task:**
>
> Thank you for recognizing the novelty of our dataset and task. We appreciate your acknowledgment of its potential contributions to the field. In our initial submission, we inadvertently omitted a detailed section on data collection. To address this, we have added a comprehensive description to the paper, as outlined below:
>
> > *"For each country, YouTube lists daily the 'most popular' videos, accessible through the YouTube Application Programming Interface (API). We collected the top 50 trending videos for each of 109 countries for 589 days between February 13, 2021 and March 17, 2023 (approximately 5,450 observations per day), resulting in a total of 3,210,050 observations and 1,302,698 unique videos. Each observation includes the unique ID of the video, the countries where it was trending, its category, title, tags, and popularity metrics, including views, likes, and dislikes. Given that the majority of videos neither go viral nor achieve significant consumption levels, gathering a representative sample of globally popular videos is inherently challenging. Previous studies collected similar datasets but over much shorter periods (1-5 months) [1,2], whereas our dataset spans more than two years with larger coverage of countries."*
>
> By including these details, we aim to highlight the dataset's uniqueness and substantial contribution to the field. We hope this additional information allows you to further appreciate the value of our work.
>
> [1] https://journals.plos.org/plosone/article?id=10.1371/journal.pone.0177865
> [2] https://journals.plos.org/plosone/article?id=10.1371/journal.pone.0278594
>
> #### **2. Insights like reach vs. popularity and ablation studies:**
>
> We are delighted that you found our insights, particularly the nuanced definitions of popularity, valuable. Conceptually, our work extends the conventional vertical notion of popularity commonly explored not only in computer science but also across social sciences. By integrating both engagement intensity and geographic reach, we propose a more comprehensive framework for understanding popularity. We believe this theoretical and conceptual innovation is a significant step forward, not only for the machine learning community but also for researchers in the web and social media as well as social science communities.
>
> #### **3. Human studies to verify video verbalization and hypothesis generation:**
>
> We appreciate your recognition of the manual evaluations as a key validation of our pipeline. In response to your feedback regarding the sample size, we expanded the evaluation from 4 to 10 videos, with each video reviewed by 30 independent evaluators, totaling 300 interactions for the evaluation of the hallucination on the video-to-text process as well as for the quality of hypothesis/explanations. The updated results remain consistent, demonstrating high reliability in addressing potential hallucinations and receiving strong ratings for the quality of hypotheses and explanations.
>
> We would like to note that our screening tests are intentionally challenging, requiring detailed attention to video content to ensure high-quality feedback. However, this does not imply that evaluators who did not pass the screening provided poor data, as the results were largely consistent across both groups. We have updated the paper to reflect these experimental changes and clarified the methodology and statistical analyses. We hope this further reinforces your positive evaluation of this aspect of our work.

---

> > ### Author Response · Authors · 2024-11-25
> >
> > ### **Weaknesses**
> >
> > #### **1. Novelty of the frame-to-text method:**
> >
> > We appreciate the reviewer’s careful attention to prior work and for highlighting relevant studies [1-4]. We acknowledge that similar frame-to-text approaches have been explored in these works, and we will revise the paper to clearly position our work as building upon these foundational studies. Additionally, we will cite these papers to ensure proper credit is given for their contributions to the field.
> >
> > While frame-to-text pipelines are not new, our work introduces several key contributions that distinguish it from prior research:
> >
> > + **A Hybrid Framework Combining Supervised Learning with LLMs:** Our framework integrates supervised learning with Large Language Models (LLMs) to enhance video popularity prediction. This hybrid approach outperforms traditional supervised methods, providing a robust and interpretable solution for complex multimodal tasks.
> >
> > + **Effective Handling of Contextual, Temporal, and Cultural Complexities:** We focus on predicting video popularity across geographic and temporal dimensions, requiring a deeper understanding of contextual and cultural factors embedded in video content. By demonstrating how LLMs capture these complexities, we extend beyond metadata-driven approaches seen in previous works.
> >
> > + **Empirical Evaluation of Prompt Strategies:** We systematically explore and evaluate the impact of different prompting techniques, such as hypothesis generation, KNN-based example retrieval, and supervised signals. This analysis provides actionable insights into leveraging prompt engineering to improve multimodal prediction tasks.
> >
> > + **Integration of Prompt Engineering and Hybrid Pipeline:** To our knowledge, the systematic combination of supervised signals, KNN-based examples, and hypothesis generation as part of a hybrid pipeline has not been explored previously. Our results demonstrate how these elements work synergistically to push the boundaries of video analysis tasks.
> >
> > + **Performance Validation through Human Evaluations:** While prior works [1-4] demonstrate the ability to generate accurate behavior descriptions, our study further validates video-to-text conversions and hypotheses through human evaluations. Importantly, we show that the hybrid framework achieves superior classification performance compared to purely supervised approaches while producing reasonable descriptions.
> >
> > To address this comment, in the revision, we aim to clarify how our work builds on the strengths of prior studies while clearly summarizing these innovations.
> >
> > [1] https://aclanthology.org/2023.emnlp-main.608/
> > [2] https://arxiv.org/abs/2309.00378
> > [3] https://arxiv.org/abs/2405.00942
> > [4] https://openreview.net/forum?id=TrKq4Wlwcz
> >
> > #### **2. Explanation of the 17k videos and metadata collection:**
> >
> > We appreciate this feedback and acknowledge that our initial submission lacked details about the data collection process. To address this, we have added a dedicated section thoroughly explaining the data characteristics and collection process, as detailed in our response to Strength 1 above.
> >
> > #### **3. Lack of negative samples and potential language biases in the dataset:**
> >
> > Regarding the lack of negative samples, as mentioned earlier in our response to the dataset strengths, collecting non-popular videos is relatively straightforward since the majority of videos do not go viral or are not even consumed. The true challenge lies in curating a representative dataset of popular videos across diverse countries and cultures. We believe that the absence of negative samples does not constitute a weakness; instead, our focus on distinguishing globally popular videos from locally popular ones addresses a more complex and nuanced task. Additionally, classifying popular vs. non-popular videos is comparatively easier than discerning global hits from local hits. By demonstrating the effectiveness of our pipeline in handling this complex task, we hope to alleviate concerns regarding the lack of negative samples.
> >
> > Regarding language bias, we acknowledge that the dataset reflects the platform's natural language distribution, which may inherently exhibit such biases. However, the sample used for our experiments closely mirrors this distribution, ensuring no specific language is significantly over- or under-represented. To enhance transparency and address this concern, we have added a new figure in the appendix (see https://drive.google.com/file/d/1zGwaQ4mkk9zysCHcG3bXD0RN_GZk4yGW). The red bars represent the language distribution across the entire dataset (i.e., the distribution of all top N videos across countries over time), while the blue ones show the language distribution specific to our experimental sample. This addition provides greater clarity on the language distributions and further strengthens the transparency of our study. Thank you for your insightful observations and for providing us the opportunity to improve this aspect of the paper!

---

> > > ### Comment · Reviewer_HVCp · 2024-11-25
> > >
> > > Thank you for your response. Please use the related work sparingly; it is very likely that more prior works exist. I have recalled a few examples that use titles, metadata, likes, views, and frame-by-frame captions. This highlights the need for a more comprehensive related works section, which would also guide relevant readers to your work. It would be more satisfactory to see a thorough literature review embedded directly into your related work section.
> > >
> > > ## Effective Handling of Contextual, Temporal, and Cultural Complexities
> > > I believe this contribution is more applicable to the dataset and orthogonal to the methodology. If it is indeed part of your methodology and I missed it, could you verify how this work improves upon "Contextual, Temporal, and Cultural Complexities"? Quantitative evaluations would help clarify this, because qualitative evaluation in lines 488–489 appears to be contradictory to this claim.
> > >
> > > ## Hybrid Framework and Pipeline
> > > Could you clarify the distinction between:
> > >
> > > > A Hybrid Framework Combining Supervised Learning with LLMs: Our framework integrates supervised learning with Large Language Models (LLMs) to enhance video popularity prediction. This hybrid approach outperforms traditional supervised methods, providing a robust and interpretable solution for complex multimodal tasks.
> > >
> > > and
> > >
> > > > Integration of Prompt Engineering and Hybrid Pipeline: To our knowledge, the systematic combination of supervised signals, KNN-based examples, and hypothesis generation as part of a hybrid pipeline has not been explored previously. Our results demonstrate how these elements work synergistically to push the boundaries of video analysis tasks.
> > >
> > > ## Explanation of the 17k Videos and Metadata Collection
> > > Please indicate where these changes are made in the manuscript.
> > >
> > > ## Language Bias
> > > Thank you for the analysis. If I understand correctly, doesn’t the graph reinforce my concern? The target class labels {"local hit", "global hit"} appear to have a confounding variable: language. As such, any model (e.g., an LLM) capable of identifying the language may face an imbalanced class distribution.
> > >
> > > For example, the probability of an English video being a global hit is 75%, whereas for other languages it is closer to 50%. This suggests two potential solutions: (1) sample the dataset to balance all languages and target classes, or (2) introduce metrics that are less sensitive to dataset imbalance, such as F-beta measures averaged across languages. Please look for other potential confounders too, these issues can seriously undermine the work.
> > >
> > > ## Negative Samples
> > > Thank you for the clarification on this point.

---

> > > > ### Author Response · Authors · 2024-11-26
> > > >
> > > > A fully revised version of the paper has been uploaded and is ready for your review.

---

> > ### Comment · Reviewer_HVCp · 2024-11-25
> >
> > Thank you for your response. The additional information is helpful, and I am somewhat inclined to improve my score. However, I am not fully satisfied with some of the points listed below. Additionally, I have a general concern: please include your proposed changes either in the main paper or at least in the appendix, and clearly state them. This will provide the reviewers with a clearer picture; otherwise, it becomes difficult to identify the exact changes.
> >
> > ### Dataset
> > To the best of my knowledge, there is no mention of the release of the dataset or supplementary code to reproduce the dataset. This significantly limits its usability. Also, did the authors explore https://kworb.net/youtube/trending_overall.html#google_vignette ? Please clearly outline your contributions in this regard.
> >
> > ### Human Study
> > I appreciate the increase in participants and videos. However, it is challenging to understand its effect without reflecting this change in the manuscript. I could not identify any revisions in the paper, could you clarify where this has been updated?
> >
> > > We would like to note that our screening tests are intentionally challenging, requiring detailed attention to video content to ensure high-quality feedback.
> >
> > Again, could you please include details of your screening test or describe it in the appendix?

---

> > > ### Author Response · Authors · 2024-11-25
> > >
> > > Thank you for your note. We have uploaded the current version of the revised manuscript (still a work in progress) with newly added content highlighted in yellow for your review. Before the deadline, we will refine the paper to ensure it meets the page limit and other requirements.
> > >
> > > Regarding the dataset, we have now clearly mentioned in the paper that, upon acceptance, we will share the dataset and video downloading code through a public repository. Given the reviewers' feedback highlighting the dataset's potential contribution to the field, we have decided to share not only the dataset used in our experiments but also the full dataset comprising 1.3M unique videos, as described in our response. The dataset will be provided in two tables: (1) a table containing video IDs with metadata for the entire dataset, and (2) a similar table with an additional column specifying the "global highly popular" vs. "local popular" labels. Alongside these datasets, we will share Python code for downloading the videos.
> > >
> > > Regarding the human validation, we have updated the relevant section. For detailed information, please refer to the revised version's survey section and the appendix, which now include the full instructions and questions used in the survey.
> > >
> > > We sincerely appreciate your prompt response and thoughtful feedback. Please let us know if you have any additional questions or suggestions!

---

> ### Author Response · Authors · 2024-11-25
>
> #### **4. Limited scope of human evaluation:**
>
> We acknowledge the concern regarding the scope of our human evaluation and have taken substantial steps to address this in the revised version of the paper. Specifically, we have significantly expanded our human evaluation process, implementing the following improvements:
>
> + **Increased Video Samples:** We expanded the number of evaluated video samples from 4 to 10.
>
> + **Expanded Participant Interactions:** Each participant reviewed 2 videos, with each video evaluated by 30 participants. This resulted in a total of 300 video-human coder interactions (about 140 interactions if considering only those who passed the screening tests).
>
> + **Rigorous Screening Criteria:** Participants were subjected to rigorous screening tests after they evaluated each video, inspecting whether they paid very careful attention to video content and detailed understanding of the material. These tests were intentionally challenging to distinguish participants from their level of engagement and deep understanding as these can be associated with the data quality.
>
> + **Result Consistency Across Groups:** The consistency in ratings across screened and unscreened participants suggests that the results are robust and reliable. While the screening criteria ensured the highest-quality feedback, participants who did not pass the tests still provided valid data, as evidenced by the overall alignment in results.
>
> We have updated the paper to reflect these improvements, including the expanded evaluation, statistical analyses, and additional clarifications. We hope these updates further strengthen the reviewer's confidence in the robustness of our human evaluation process and its contribution to validating the quality of our findings.
>
> #### **5. Limited technical contributions:**
>
> We appreciate the reviewer's feedback on the technical novelty of our work. While it is true that our framework leverages existing prompting strategies, its novelty lies in the careful integration and application of these techniques to a novel task. Specifically, our approach demonstrates how sequential prompting, hypothesis generation, and supervised signals work synergistically to achieve state-of-the-art results while providing interpretable outputs.
>
> Our extensive ablation studies systematically evaluate these components both individually and in combination, highlighting their unique contributions to the framework's performance. This comprehensive analysis not only benchmarks existing techniques but also provides valuable insights into their impact on multimodal LLM-based tasks, establishing a basis for future research in this domain.
>
> To further strengthen our contribution, we conducted additional experiments using an advanced Vision-Language Large Model (VLLM), Gemini. Specifically, we used the Gemini 1.5 Pro model, accessed as API in Vertex library. The input to the model included a combination of the summariser prompt and the final prediction prompt. The sequential frames were aligned with the video’s existing captions to ensure that visual and verbal elements were synchronized before being fed into the LLM to generate the video-to-frame summary for the entire video. This process closely follows the steps described in Section 3.2 where “Frame Extraction” step (extracting 5 frames per minute) and the “Caption Matching and Data Integration” step were employed to align captions and frames for LLM processing. Subsequently, "Frames to Text Conversion and Summarization" step generated the final video summary using an LLM call. For prediction, we used the same final prompt detailed in Figure A2, ensuring consistency with our primary method.
>
> These experiments confirmed that our strategy—incorporating sequential prompting, hypothesis generation, and supervised signals—consistently improves prediction performance, even with this state-of-the-art model. This underscores the generalizability and robustness of our approach across different model architectures. The results of these experiments, along with a detailed figure, are included in the appendix (see https://drive.google.com/file/d/1pZjUr5SHD9a0Y-nPczr6adwxvJWzfVSI).
>
> We believe these efforts collectively highlight the significance of our work in advancing multimodal LLM research and its applications. Our approach is particularly impactful for tasks requiring a nuanced understanding of broad contextual and cultural information beyond what is readily available or visible on the platform, as demonstrated in video popularity prediction.

---

> ### Author Response · Authors · 2024-11-25
>
> Yes, we will ensure the literature review section is improved in the final revision by the deadline. Thank you for your insightful feedback!
>
> Regarding your comment on "Contextual, Temporal, and Cultural Complexities," we recognize that "temporal complexity" is not addressed in our tasks, though it is encoded in our dataset, as you pointed out. To avoid confusion, we will remove the term when describing the pipeline's capabilities. However, we believe the pipeline effectively captures contextual and cultural complexities, as demonstrated in the first two examples in Figure 2. The negative example you mentioned was intentionally included to highlight potential areas for improvement. We hope this clarifies our approach.
>
> We tend to use the term "hybrid framework" when describing the integration of supervised signals into our pipeline. Upon reflection, the term "hybrid" might seem ambiguous in this context. We will revise and clarify explanations without using such an ambiguous term throughout the paper. As an example, we have revised our contribution accordingly (see lines 80-92).
>
> To address the dataset comparison, we have highlighted changes regarding the dataset in yellow in the paper. Additionally, we note that, to the best of our knowledge, the dataset provided by https://kworb.net/youtube/trending_overall.html is significantly limited in scope and usability, especially compared to our dataset. While it provides "last 24-hour" trending data or "aggregated" top lists per year or country, it lacks detailed information about a video's geographic spread or its evolution across countries. In contrast, our dataset offers longitudinal daily snapshots, capturing a video's full trajectory from its first appearance on the trending list to its exit from the list. This comprehensiveness enables more accurate classification into categories such as global big hits and local hits. We hope this distinction clarifies the differences and further underscores the unique value of our dataset.
>
> Lastly, we sincerely appreciate your thoughtful suggestions regarding the language imbalance issue. If the paper is accepted, we will incorporate these proposed solutions into the camera-ready version. In the meantime, we would like to assure you that language imbalance is unlikely to pose a significant issue in our study. For example, English accounts for approximately 40% of the global big hit category (not 75%) and 15% of the local hit category. Additionally, our analysis of prediction accuracy across languages indicates that prediction performance is not simply a function of the representation of each language in the dataset and is comparable across major and minor languages (see https://drive.google.com/file/d/1oYfo0Dsl_oidG4kbN3qnJkDoIoVdLqVf). These findings suggest that language imbalance does not act as a confounding factor in our predictions. We will include these analyses and clarifications in the revised version to enhance transparency and address this concern.

---

> ### Comment · Reviewer_HVCp · 2024-12-02
>
> Thank you for the changes, I would again urge the authors to improve the literature review, it seems limited at the moment.
>
> > Building on these innovations, our approach introduces a frame-to-text  transformation pipeline that
> converts video frames into sequential textual descriptions via VLMs.
>
> There are multiple such works, https://openaccess.thecvf.com/content/CVPR2024/papers/Zhang_MM-Narrator_Narrating_Long-form_Videos_with_Multimodal_In-Context_Learning_CVPR_2024_paper.pdf and [2] https://arxiv.org/abs/2309.00378
> [3] https://arxiv.org/abs/2405.00942.
>
> Thank you for the changes, the presentation and my overall understanding of the paper has improved.

---

### Meta-Review · Area_Chair_Hquy · 2024-12-12

**Metareview:**

The paper tackles video popularity prediction. The reviewers praise the proposed dataset but question novelty (wrt prior work that uses video metadata and ASR), limited human evaluation and comparison to other methods, aspects of the method (e.g. measurements used), and clarity. Concerns are not sufficiently well addressed in the rebuttal, resulting in two negative and one weakly positive final ratings.

**Additional Comments On Reviewer Discussion:**

Two of the three reviewers meaningfully engaged in the discussion with the authors

---

### Decision · Program_Chairs · 2025-01-22

Reject